# Spatiotemporal organization of membrane protein controls bacterial extracellular electron transfer

Youngchan Park [1,3], Tianlei Yan [1], Zhiheng Zhao[1], Bing Fu[1,4], Muwen Yang [1,5], Farshid Salimijazi [2,6], Buz Barstow [2] & Peng Chen [1] ✉

Extracellular electron transfer (EET) is essential for electroactive microbes' physiology and biotechnological applications. Many such microbes are Gram-negative bacteria, in which EET must cross two membranes and the periplasm, necessitating spatial and temporal collaborations of various EET proteins that reside at different cellular compartments, for which little is known. Using single-molecule/single-cell-level fluorescence microscopy and electrochemical manipulations, we discover that in the electroactive bacterium *Shewanella oneidensis*, the inner-membrane electron-transfer hub protein CymA undergoes spatial reorganization into localized regions during active EET with dispersed formation dynamics, subsequently driving the colocalization of its direct electron-transfer partners in the periplasm. Correlated single-cell-level photoelectrochemistry-fluorescence microscopy further proves the critical function of CymA reorganization in enabling EET. A multitude of evidence suggests that CymA reorganization stems from biomolecular condensate formation, likely initiated by association with menaquinone-rich inner-membrane domains. These orchestrated spatiotemporal protein dynamics extend the functional roles of biomolecular condensates to include facilitation of EET in bacteria, with broader implications for cellular processes.

Extracellular electron transfer (EET) is the process that electroactive microbes use to exchange electrons with extracellular electron acceptors or donors[1,2]. Beyond its foundational importance for these microbes' anaerobic respiration and interspecies electron transfer, EET finds applications in diverse realms, including in renewable energy technologies (e.g., microbial fuel cells, electrosynthesis, electrolysis)[3–6] and environmental remediation of toxic metals and organic pollutants[7,8]. Understanding the mechanisms of EET is thus crucial, not only for advancing fundamental microbiology but also for gaining knowledge to engineer the associated cellular processes for applications.

Many such electroactive microbes are Gram-negative bacteria, in which the EET must cross two membranes (inner and outer

membranes, IM and OM) and the periplasm of the cell envelope, involving intricate collaborations of numerous cellular components that reside in different cellular compartments. *Shewanella oneidensis* MR-1, a model electroactive bacterium, uses >40 different cytochrome proteins[9,10] within the cell envelope to carry out outward (outflow) and inward (uptake) EET (Fig. 1a). In particular, to exchange electrons with extracellular solids, soluble metal ions, or electrodes, the EET pathway in *Shewanella* commissions a tetra-heme type *c* cytochrome, CymA, as a central electron-transfer hub anchored on the IM (Fig. 1a)[11]. It exchanges electrons with menaquinone, an anaerobic electron transport chain component in the IM[12], which can obtain electrons from the IM-anchored *D*-lactate dehydrogenase (Dld-II)[13] and anaerobic NADH

[1]Department of Chemistry and Chemical Biology, Cornell University, Ithaca, NY, USA. [2]Department of Biological and Environmental Engineering, Cornell University, Ithaca, NY, USA. [3]Present address: Department of Chemistry, Indiana University, Bloomington, IN, USA. [4]Present address: Department of Biomedical Engineering, City University of Hong Kong, Hong Kong, China. [5]Present address: L.E.K. Consulting, Boston, MA, USA. [6]Present address: International Flavors & Fragrances, Palo Alto, CA, USA. ✉e-mail: pc252@cornell.edu

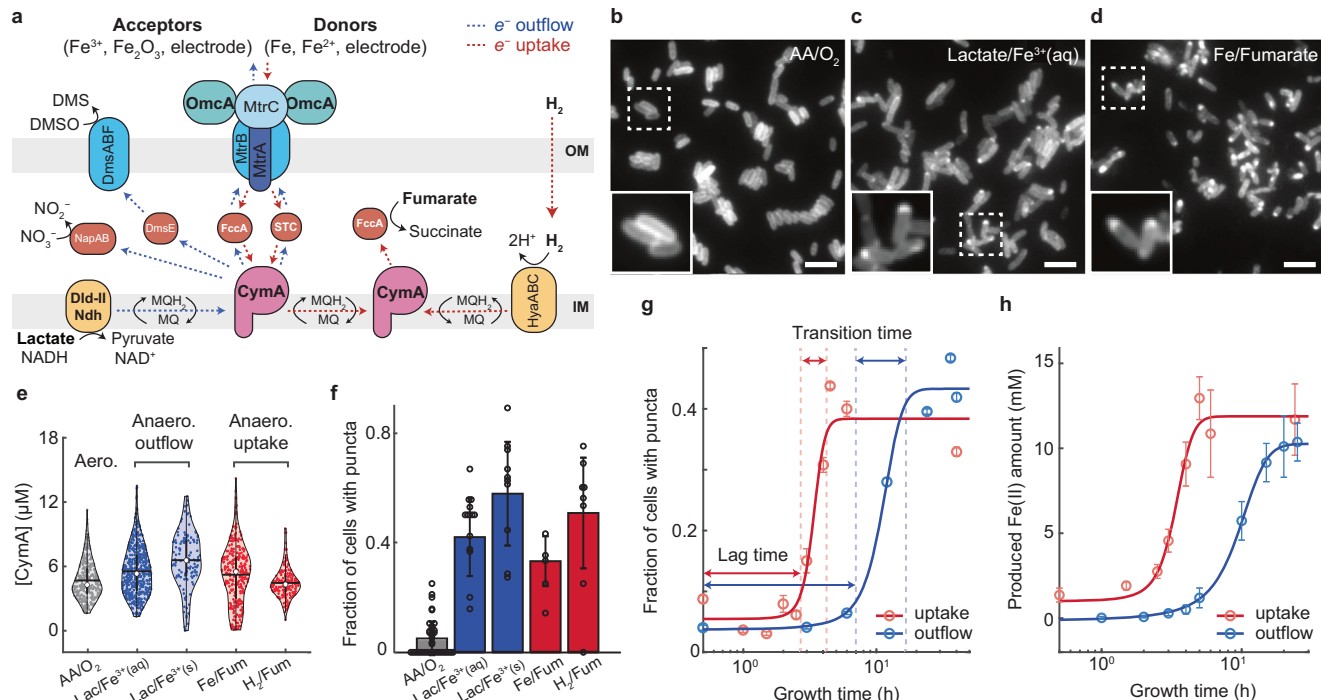

**Fig. 1 | CymA spatially reorganizes into punctate patterns during EET under anaerobic growth. a** Major EET pathways in *Shewanella oneidensis*. Dotted arrows indicate outflow (blue) and uptake (red) EET directions. MQ/MQH₂: oxidized/reduced menaquinone. **b-d** Wide-field fluorescence images of ensemble-photoactivated CymA^PmC in *S. oneidensis* grown under aerobic (**b**) or anaerobic (**c, d**) conditions with different electron donor/terminal electron acceptor pairs for outflow (**c**, lactate/ferric citrate) or uptake (**d**, metal iron/fumarate) EET. Insets: zoom-in of dotted boxes. Scale bars: 5 µm. **e** Cellular concentrations of CymA^PmC grown aerobically (column 1) or anaerobically with different electron donor/acceptor pairs for outflow (columns 2–3) or uptake (columns 4–5) EET. AA: amino acids; Lac: lactate; Fe³⁺(aq): ferric citrate; Fe³⁺(s): ferric oxide; Fe: metal iron; Fum: fumarate. White circles: median; horizontal black lines: mean; vertical black lines: first and third quartiles. Data collected from 2 to 3 biological replicates. **f** Fraction of cells showing CymA^PmC puncta after 2-day aerobic or anaerobic growth with

outflow/uptake EET. Open circles: fractions of cells with puncta from individual images. Means and error bars (s.d.) are from multiple images from 2 to 3 biological replicates, with the calculations weighed by the cell counts. **g** Fraction of cells exhibiting CymA^PmC puncta vs. time under anaerobic growth with outflow (blue circles, lactate/ferric citrate as the redox pair) and uptake (red circles, metal iron/fumarate as the redox pair) EET. Different time points are from different sets of cells. Lines: fits with a sigmoidal function (Supplementary Notes 4). Means and error bars (s.e.m.) are from multiple images from 1-3 biological replicates, with the calculations weighed by the cell counts. **h** Produced Fe(II) concentration in the medium vs. time under anaerobic growth with outflow (blue circles, lactate/ferric citrate as the redox pair) and uptake (red circles, metal iron/fumarate as the redox pair) EET. Fe(II) amount is measured using ferrozine assay (Supplementary Methods 1.4). Means and error bars (s.d.) are from 3 biological replicates. Lines: fits with sigmoidal functions as in (**g**). Source data are provided in a Source Data file.

dehydrogenase (Ndh)[14]. In the periplasm, CymA exchanges electrons with STC, a small tetra-heme cytochrome, and FccA, flavocytochrome *c*3, which also functions as a fumarate reductase[15,16]. Both STC and FccA can shuttle electrons toward the OM cytochrome complex MtrABC/OmcA, which serves as a metal reductase. Electron transfer through these cytochromes are dominated by transient interactions instead of through stabilized complexes[15]. The central electron-transfer hub role of CymA also underlies *Shewanella*'s ability to detoxify nitrates and organic pollutants like DMSO (Fig. 1a)[17,18].

Many studies have provided insights into the structure and function of EET proteins individually[19,20]; yet, little is known about how the cell orchestrates the spatial and temporal actions of the EET proteins that reside in different cellular compartments. However, their spatiotemporal coordination is critical for biological electron transfer, which relies on the close proximity of electron-transfer partners, typically achieved through specific molecular interactions[21]. Here, we report a single-cell/single-molecule-level imaging study of EET proteins in *Shewanella oneidensis*. We discover that the IM electron-transfer hub protein CymA undergoes spatial reorganization into localized regions without discernible protein upregulation when cells are actively engaged in EET. Such spatial reorganization occurs irrespective of the EET direction, the specific terminal redox pairs, or the other redox partners in the pathway; it also drives the colocalization

of CymA's direct EET partners in the periplasm and is likely initiated by the association with menaquinone-rich membrane domains, both of which should facilitate EET across the cell envelope. In situ single-cell-level electrochemical fluorescence microscopy further unveils that the seemingly slow spatial reorganization of CymA in population analysis does not reflect a slow, gradual formation process, but arises from the dispersed reorganization dynamics among individual cells. Correlated single-cell photoelectrochemistry-fluorescence microscopy further proves cells' higher EET activity after spatial reorganization, demonstrating its functional roles. Single-molecule tracking and imaging analysis reveal that such localized spatial organizations exhibit liquid-like properties, such as spatially condensed structures, reversible formation comprising both temporal emergence and dissipation, molecular mobility, and dynamic exchange and coexistence with external CymA populations. These properties suggest that CymA reorganization likely stems from biomolecular condensate formation, which can enhance electron changes with menaquinone in IM. Altogether, our findings provide insights into how cells coordinate the spatiotemporal actions of proteins that reside in different cellular compartments to control EET and may extend the functional roles of biomolecular condensates to include facilitation of EET in bacteria, suggesting their broader relevance in cellular function.

## Results

### Inner-membrane electron-transfer hub protein CymA forms puncta in cells for EET

To investigate EET in *S. oneidensis*, we started with CymA, the IM central electron-transfer hub in anaerobic respiration (Fig. 1a). To image CymA's spatiotemporal behaviors in live cells under physiological expression, we tagged its C-terminus with the photoactivable red fluorescent protein PAmCherry1 (PmC) at its chromosomal locus, creating CymA$^{PmC}$. The PmC tag enables conventional fluorescence imaging after photoactivation, as well as single-molecule localization−based tracking and super-resolution imaging. The tagged CymA$^{PmC}$ proved to be >92% intact in the cell and as functional as the wild-type in iron reductivity (Supplementary Methods; Supplementary Notes 1).

We first grew *S. oneidensis* cells aerobically, under which CymA is known to not be engaged in EET[11,22]; CymA$^{PmC}$ shows a homogeneous distribution over the cell envelope (Fig. 1b), consistent with being an IM-anchored protein. We then grew the cells anaerobically with lactate and ferric citrate as the sole electron donor and terminal acceptor, under which the cells are known to undertake outflow EET via the Mtr complex passing through CymA (Fig. 1a, blue dotted lines). CymA$^{PmC}$ formed spatially localized puncta of ~400–600 nm in size under diffraction-limited imaging (Fig. 1c; note we employed machine learning algorithms[23] to identify CymA$^{PmC}$ punctate pattern in the cell; Supplementary Notes 2). Note we exposed the anaerobically grown cells to air for sufficient PmC maturation[24] prior to fluorescence imaging, and this exposure to air is not the cause of punctum formation (Supplementary Notes 3). We also examined cells anaerobically grown with metal iron/fumarate as the electron donor/acceptor pair, where cells engage in EET through the same pathway but in the uptake direction[25] (Fig. 1a, red dotted lines); CymA$^{PmC}$ exhibited the same punctate features (Fig. 1d). We further imaged CymA$^{PmC}$ in cells grown anaerobically with other electron donor/acceptor pairs, such as lactate/ferric oxide ($e^-$ outflowing via Mtr with insoluble terminal acceptor) or $H_2$/fumarate ($e^-$ uptaking via hydrogenase) (Fig. 1a and Supplementary Table 1). CymA$^{PmC}$ consistently showed punctate features in the cell as long as it was actively engaged in EET, regardless of the EET direction, the specific terminal redox pairs (and their being soluble or insoluble), the other redox partners in the pathway, or the fluorescent protein tag (i.e., tagged with PmC and GFP; Fig. 3b later). Notably, the puncta were predominantly located near the cell poles (Supplementary Fig. 20).

To probe whether CymA's spatial reorganization is coupled to changes in its expression level, we quantified CymA$^{PmC}$'s cellular concentrations (Supplementary Methods 1.5). CymA appears to be constitutively expressed, consistent with previous transcriptional analysis[26,27] and showing no significant changes regardless of its active participation in EET or which external redox pairs were used (~5.3 µM in concentration, ~4.0 × 10³ in copy number, on average; Fig. 1e).

Moreover, our imaging unveiled heterogeneity in punctum formation among individual cells (Fig. 1c, d and Supplementary Fig. 21). After 2 days of anaerobic growth, 40–50% of cells formed puncta and this fraction remained largely unaffected by the EET direction and the specific redox partners involved, compared with the insignificant fraction (<5%) in cells grown aerobically (Fig. 1f).

To probe whether the spatial reorganization of CymA into punctate patterns is coupled temporally to *S. oneidensis*'s gain in EET activity, we imaged cells over different periods of anaerobic growth with lactate/ferric citrate as the redox pair for outflow EET. CymA puncta emerged in a population of cells, following a sigmoidal profile with a lag time of $7 \pm 1\,h$ and a transition time of $8 \pm 5\,h$ (Fig. 1g, blue). In contrast, when EET was reversed to the uptake direction using the metal iron/fumarate redox pair, the emergence of the punctate pattern showed much shorter lag and transition times, at $2.7 \pm 0.2$ and $1.4 \pm 0.6\,h$, respectively (Fig. 1g, red). More strikingly, the two lag times of CymA's punctate pattern emergence coincide with the emergence

of outflow and uptake EET activity of cells at $5 \pm 1\,h$ and $2.3 \pm 0.2\,h$, respectively, measured by the amount of Fe(II) produced in the medium from ferric reduction or metallic iron oxidation (Fig. 1h and Supplementary Methods 1.4), which also agree with previous biochemical assays ($6 \pm 0.1$ and $1.3 \pm 0.1\,h$, respectively, Supplementary Fig. 22)[22,28]. Therefore, the spatial reorganization of CymA into puncta appears crucial for the cell's EET.

### Single-cell dynamics of CymA spatial reorganization

Our analysis of cell populations over various anaerobic growth periods revealed heterogeneity of CymA's spatial reorganization into puncta among individual cells (Fig. 1c, d), and these puncta emerge over a timescale of many hours (Fig. 1g). However, such analysis does not inform that in a single cell, whether such puncta form gradually or appear more suddenly with variable delay times among individual cells.

To monitor punctum formation dynamics at the single-cell level, we imaged CymA$^{PmC}$ in cells deposited on a transparent indium-tin-oxide (ITO) electrode in a microfluidic electrochemical cell (Fig. 2a(i)). Under anaerobic conditions and upon applying electrochemical potential, *S. oneidensis* is known to be able to directly exchange electrons with electrodes, in which CymA, and the Mtr complex, are key EET players[29,30] (Fig. 1a). An alternative approach would be to image cells in situ while exchanging growth condition from aerobic to anaerobic; but anaerobic growth conditions typically either use solids (e.g., metal iron or ferric oxide) or generate precipitates (e.g., the ferrous citrate precipitate from ferric citrate reduction), hindering fluorescence imaging. Electrochemical microscopy approach, as employed here, also offers ease of tuning the electron-transfer driving force. Moreover, we chose to focus on cathodic uptake EET condition, in which CymA puncta are expected to emerge quicker (a few hours; Fig. 1g, red circles), facilitating in situ fluorescence microscopy, instead of anodic outflow EET direction, under which puncta formation is much slower (lag time: $9 \pm 1\,h$; Supplementary Fig. 23) and photobleaching of fluorescent protein tag substantially limits our observation time. Although EET uptake in *S. oneidensis* is less studied and its physiological relevance is less established, it is as important as EET outflow for biotechnological applications (e.g., for microbial electrosynthesis[31,32]).

We cultured the cells first aerobically for 2 days before placing them in the anaerobic electrochemical cell and photoactivating all cellular CymA$^{PmC}$ for imaging, where the fluorescent protein tag (PmC) expressed during aerobic culture was sufficiently matured (Supplementary Methods 1.6). The initial distributions of CymA$^{PmC}$ in cells are mostly uniform expectedly (Fig. 2b, left). We then applied a cathodic potential of $-0.48\,V$ (vs. Ag/AgCl), sufficient to drive electron flow into the cells (Supplementary Fig. 24 on redox potentials of proteins in Mtr pathway[29,33]). Fumarate in an anaerobic minimal medium was supplied continuously to be a terminal electron acceptor. We imaged the photoactivated CymA$^{PmC}$ in each cell over 6 h. Most CymA$^{PmC}$ puncta were observed in cells within 4 h (Fig. 2f, >90%) during which there was no substantial synthesis of CymA$^{PmC}$ in the cell (Supplementary Notes 3). After 6 h, CymA$^{PmC}$ puncta were observed in many cells (Fig. 2b, right, and Supplementary Fig. 25a, b). Without applying potential, cells did not form puncta (Supplementary Fig. 25c), indicating that punctum formation is indeed driven by EET, rather than by mere contact with the electrode or the nutrient-limited solution conditions. Remarkably, individual cells exhibited heterogeneous temporal behaviors in punctum formation, often with a noticeable time lag before the puncta appeared more suddenly (Fig. 2c).

To quantify the temporal dynamics of CymA punctum formation, we computed the time-dependent skewness, $\widetilde{\mu}_3$, of the pixel fluorescence intensity distribution in each cell. As CymA$^{PmC}$ reorganizes into puncta, such fluorescence intensity distribution shifts from a symmetric shape ($\widetilde{\mu}_3 = 0$) to a skewed shape with a prolonged tail

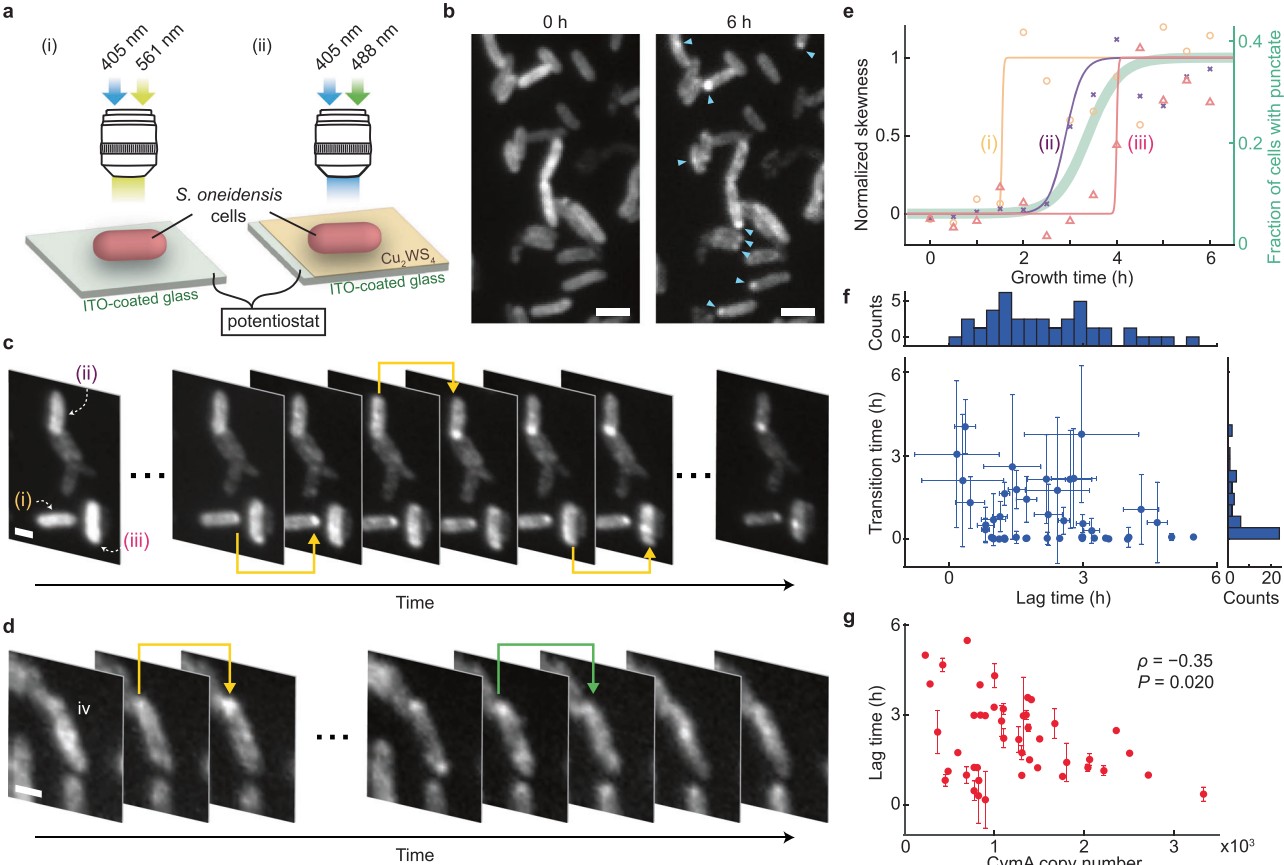

**Fig. 2 | CymA punctum forms quickly in individual cells but with highly dispersed lag times. a** Electrochemical (i) and photoelectrochemical (ii) fluorescence microscopy of single cells, integrating a (photo)electrochemical microfluidic cell on an epi-fluorescence microscope. ITO, connected to a potentiostat, serves as the working electrode. Photoactivation of PmC and/or excitation of $Cu_2WS_4$ semiconductor is by a 405-nm laser; excitation of PmC red fluorescence by a 561-nm laser; GFP by a 488-nm laser. More details in Supplementary Methods 1.6 and 1.9. **b** Fluorescence images of ensemble-photoactivated $CymA^{PmC}$ in cells initially grown aerobically and then placed anaerobically on ITO substrate, captured at 0 h (left) and 6 h (right) after applying electrochemical potential at −0.48 V (vs. Ag/AgCl). The corresponding chronoamperometric data in Supplementary Fig. 27. Scale bars: 2 µm. Arrowheads: locations of puncta. These are representative images from 10 repeated experiments. **c, d** Fluorescence image sequences of exemplary cells as in (**b**), taken every 30 min till 6 h. Yellow arrows: transition point where cells

(numbered i, ii, iii) form puncta; green arrow: transition point where a punctum dissipates in cell no. iv. Scale bars: 1 µm. The image contrast in (**b**–**d**) is the same throughout the respective time course. **e** Normalized skewness of fluorescence pixel intensity distribution in three cells from (**c**) vs. time (colored symbols). Colored lines: sigmodal fits. Thick green line: Fraction of cells showing puncta vs. anaerobic growth time with uptake EET (right *y*-axis; same as the red line in Fig. 1g). **f** Correlation between lag time and transition time for CymA punctum formation in single cells (*n* = 45). The measure of centre and the error bars: the fitted values and the standard error from sigmoidal fitting. Top/bottom: projected histograms. **g** Correlation between cellular CymA copy number and lag time from single cells (*n* = 45). The measure of centre and the error bars: the fitted values and the standard error from sigmoidal fitting. *ρ*: Pearson's correlation coefficient; *P*-value is calculated by two-sided *t*-test (*P* < 0.05, significant). Source data are provided in a Source Data file.

at the higher intensity end ($\widetilde{\mu}_3 > 0$; Supplementary Notes 5). This skewness can effectively identify cells with puncta, consistent with assignments by machine learning algorithms (Supplementary Notes 5 and Supplementary Fig. 9d). More importantly, skewness-vs-time trajectories revealed distinctive temporal behaviors of punctum formation in individual cells, showing clear time lags followed by more sudden increases from punctum formation (Fig. 2e, i, ii vs. iii). Notably, in a minor fraction of cells (~10%), the CymA punctum can dissipate (Fig. 2d(iv)), indicating the *reversibility* of punctum formation.

We fitted the skewness-vs-time trajectories of individual cells with a sigmoidal function (Fig. 2e), as we did for bulk-level observations (Fig. 1g), allowing for extracting a lag time and a transition time, which quantify the time lag prior to punctum formation and the rate of punctum formation, respectively (Supplementary Notes 5). Remarkably, individual cells differ widely in their lag times (Fig. 2f, upper), with a median of 2.2 h (interquartile range (IQR) = 1.1 – 3.0 h) that is comparable to that from ensemble observation (~2.7 h; Fig. 2e, green).

Transition times also vary substantially among individual cells (Fig. 2f, right); over half of the cells exhibit transition times <0.5 h, much shorter than that from ensemble observation (~1.4 h; Fig. 2e, green). No clear correlation exists between single-cell lag time and transition time (Fig. 2f); we note that this lack of correlation could partly result from the limited temporal resolution of our imaging in capturing the shorter transition times. In contrast, there is a robust negative correlation between the cell's CymA copy number and lag time (Fig. 2g), suggesting that more CymA copies facilitate the initiation of its spatial reorganization, probably resulting from easier encountering among CymA molecules in the cell. Furthermore, applying a potential with a lower electron-transfer driving force (−0.28 V vs. Ag/AgCl) led to similar results (Supplementary Fig. 26).

Altogether, these single-cell behaviors reveal that the seemingly slow emergence of CymA puncta observed in bulk analysis (Fig. 1g) is not indicative of a slow, gradual formation but rather stems from the dispersion of lag times among individual cells. The actual punctum formation, quantified by the transition time, is much faster.

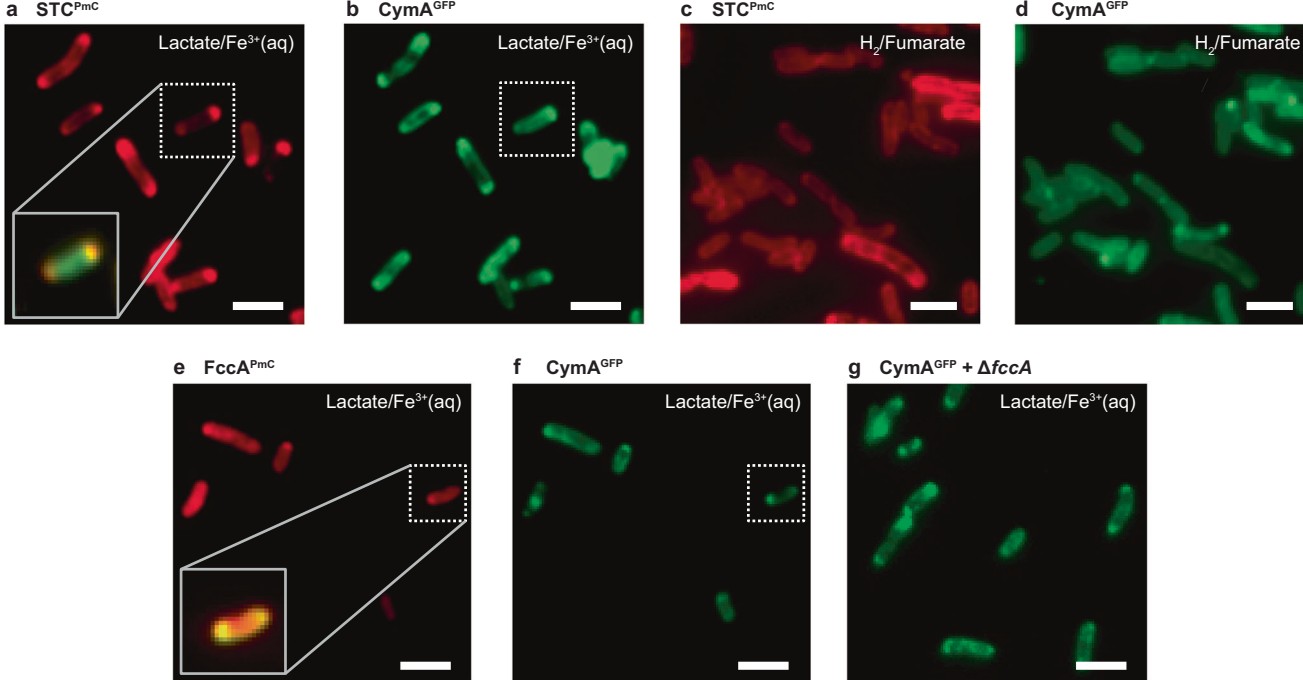

**Fig. 3 | CymA punctum formation is spatiotemporally coordinated with its direct EET partners. a,b** Fluorescence images of ensemble-photoactivated STC[PmC] (**a**, red channel) and CymA[GFP] (**b**, green channel) in the same cells grown anaerobically with lactate/ferric citrate as the electron donor/acceptor pair for outflow EET. Same growth condition for (**e–g**). Inset: overlaid zoom-in image of STC[PmC] and CymA[GFP] in dotted boxes. **c, d** Fluorescence images of ensemble-photoactivated STC[PmC] (**c**) or CymA[GFP] (**d**) in cells grown anaerobically with hydrogen/fumarate as the electron donor/acceptor pair, where EET does not involve STC. **e, f** Fluorescence images of ensemble-photoactivated FccA[PmC] (**e**, red channel) and CymA[GFP] (**f**, green channel) in the same cells. Inset: overlaid zoom-in image of FccA[PmC] and CymA[GFP] in dotted boxes. **g** Fluorescence image of CymA[GFP] in Δ*fccA* cells. All scale bars: 5 μm. All the experiments were repeated 3 times.

## Spatiotemporal coordination of CymA reorganization with cellular EET partners

Given that CymA punctum is indispensable in EET and forms dynamically, and considering that biological electron-transfer generally requires partners to be in proximity[21], we hypothesized that the punctum formation is to localize CymA spatiotemporally to cooperate with its EET partners. Under our anaerobic growth conditions (i.e., using lactate/ferric citrate, lactate/ferric oxide, metal iron/fumarate, hydrogen/fumarate, or cathode/fumarate as terminal electron donor/acceptor pairs), the EET pathways in which CymA is the central hub comprise many proteins across the cell envelope (Fig. 1a). On the OM, the protein complex MtrABC/OmcA exchanges electrons directly with extracellular redox components; imaging PmC-tagged OmcA did not observe punctum formation during EET (Supplementary Notes 6.1), consistent with previous electron microscopy studies[34]. *S. oneidensis* is also known to form OM extensions at localized spots on the outer membrane for long-distance electron transfer[35]; by staining the membrane with a dye, we found no significant colocalization of such OM extensions with CymA puncta under EET (Supplementary Notes 6.2). We further imaged PmC-tagged Dld-II and Ndh, both on the IM and part of the EET pathways (Fig. 1a); no punctate spatial pattern was observable in either (Supplementary Notes 6.1). Therefore, none of these OM and IM proteins colocalizes with CymA during EET, ruling out their being the reason for CymA punctum formation; these results are perhaps unsurprising, as neither of these proteins interacts *directly* with CymA.

In the periplasm, the tetra-heme cytochrome STC and the flavo-cytochrome FccA are known to interact directly with CymA to shuttle electrons to/from MtrA as part of EET[15] (Fig. 1a). We made chromosomally tagged, functional STC[PmC] or FccA[PmC] in the cell (Supplementary Notes 1). In aerobically cultured cells, STC[PmC] exhibits a homogeneous distribution on the cell envelope with an average copy number of ~600

(Supplementary Fig. 28). Strikingly, under anaerobic growth with outflow EET (i.e., with lactate/ferric citrate as redox pairs), STC[PmC] forms clear punctate patterns as CymA, with a ~70% increase in copy number (Fig. 3a and Supplementary Fig. 28). Double-tagging in the same cell (i.e., STC[PmC] + CymA[GFP]) and two-color fluorescence imaging further confirmed their colocalization (Fig. 3a inset and b; Supplementary Methods 1.5). We further cultured the cells anaerobically with hydrogen and fumarate; here, hydrogen can diffuse into the cell and get oxidized by IM-anchored hydrogenase to supply electrons to CymA and then to fumarate, under which STC is not engaged in EET (Fig. 1a). Remarkably, STC[PmC] maintains a homogenous spatial distribution (Fig. 3c), while CymA[GFP] still formed puncta (Fig. 3d and Supplementary Fig. 29), indicating that STC is not the driver of CymA punctum formation. Therefore, as long as CymA engages in electron transfer across the IM for anaerobic respiration (i.e., not necessarily through the entire envelope), CymA spatially reorganizes into localized puncta.

Similar experiments with FccA[PmC] revealed its significant increase in copy number by ~300% from aerobic to anaerobic growth (Supplementary Fig. 28), consistent with a previous transcriptional analysis[27,36]. More importantly, FccA[PmC] forms clear punctate patterns upon engaging in active EET (Fig. 3e). Colocalization of FccA with CymA puncta was again confirmed by double-tagging in the same cell (FccA[PmC] + CymA[GFP]) (Fig. 3e, f). We further deleted *fccA* (here STC can compensate for the electron shuttling role; Supplementary Notes 1), CymA[PmC] still formed puncta under anaerobic EET (Fig. 3g), ruling out FccA driving CymA punctum formation. Therefore, under anaerobic respiration, CymA forms puncta, which subsequently drives spatial colocalization of its direct electron-transfer partners STC and/or FccA in the periplasm. Although FccA and STC may not independently form condensates or mix into CymA condensates, their spatial proximity to CymA puncta is mediated through direct protein-protein interactions[15,16]. This interaction-driven colocalization effectively

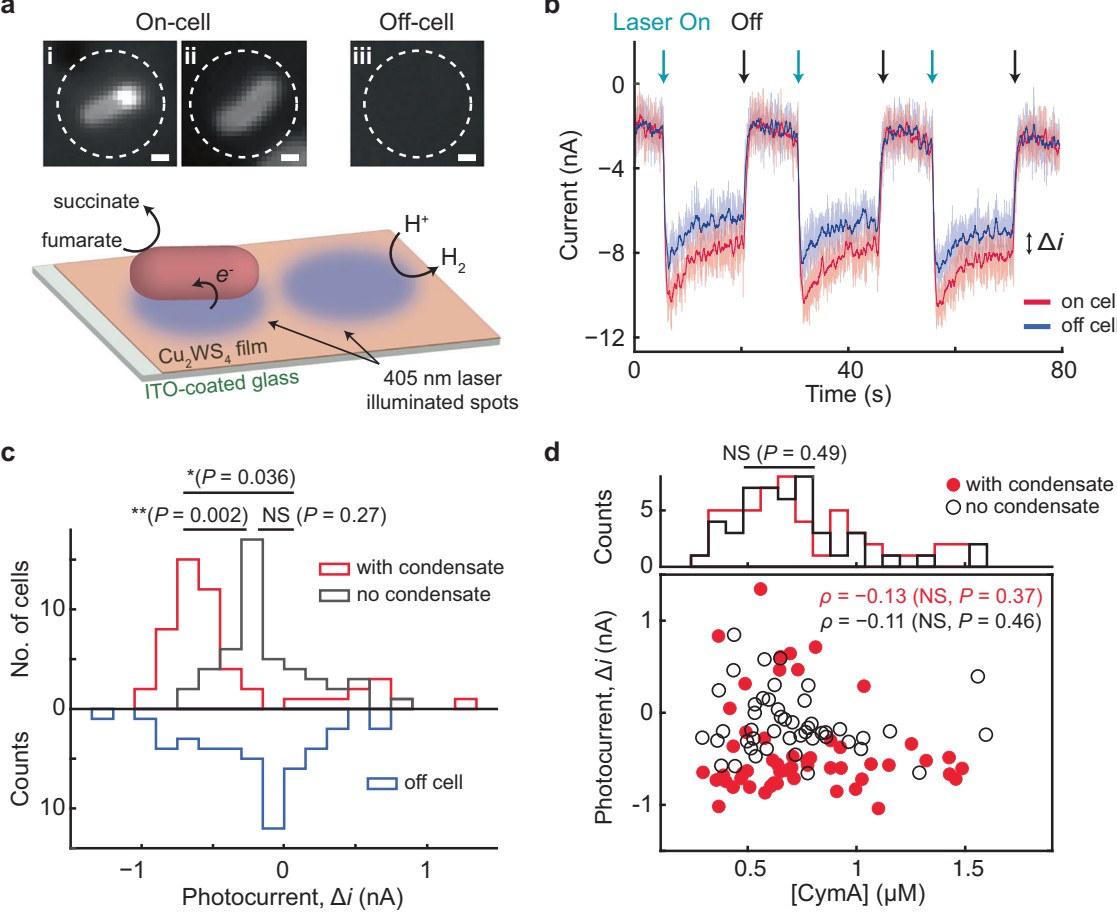

**Fig. 4 | CymA punctum formation renders higher EET flux in single cells.**
**a** Lower: illustration of single-cell photoelectrochemical current measurement. Cells were placed on top of a $Cu_2WS_4$ thin-film photoelectrode atop of ITO as in Fig. 2a(ii). A 405-nm laser excites charge carriers in $Cu_2WS_4$, illuminating a single cell (left) or off the cell nearby (right). Detailed diagram in Supplementary Fig. 1c. Top: exemplary fluorescence images of single CymA^GFP cells that formed puncta (i) or had not (ii) on a $Cu_2WS_4$ thin-film photoelectrode, after 2-day anaerobic growth with lactate and ferric citrate, along with the image of a representative off-cell location (iii). Dotted circles indicate the size and position of the 405 nm laser spot. Scale bars: 0.5 μm. **b** Representative time trajectories of local electrochemical current when the 405 nm light (198 W/cm²) is placed on a cell with a punctum (red) or at a nearby off-cell location (blue) atop of a thin-film $Cu_2WS_4$ photoelectrode

during on-off light modulation, at −0.20 V (vs. Ag/AgCl). Background dark current is subtracted. Solid colored lines: 20-point moving-average smoothed of raw data (faint-colored). Cell-induced photocurrent $\Delta i$ is defined as the steady-state current difference between on and off cells during laser on periods. **c** Histograms of single-cell $\Delta i$ with CymA puncta (red, $n = 53$) or without (gray, $n = 47$), along with control measurements on $Cu_2WS_4$ thin-film without cells (blue, $n = 48$). Statistical significance was determined using Welch's two-sample t-test (NS, not significant; $P < 0.05$, significant). **d** Correlation between cellular CymA concentration and photocurrent of single cells. Circles: individual cells containing puncta (solid red) or not (open black). Denoted are Pearson's correlation coefficient $\rho$ for cells with puncta (red) or without (black), and their $P$ values (two-sided t-test: $P < 0.05$, significant). Top: Projected histogram. Source data are provided in a Source Data file.

brings the electron transfer partners into proximity, facilitating electron exchange for efficient EET.

## Spatial reorganization renders higher EET flux in single cells
The above results show that CymA puncta form dynamically for EET in *S. oneidensis*; this reorganization subsequently drives the spatial colocalization of CymA's direct electron-transfer partners, STC and FccA, which leads to geometric proximity and are thus expected to facilitate EET. To directly probe the impact of CymA puncta on EET efficiency, we measured cells' electron uptake capability using photoelectrochemical microscopy in correlation with fluorescence microscopy at the single-cell level (see Supplementary Notes 7 and 8 about complications on measuring electron outflow via photoelectrochemistry and the consideration regarding potential reactive oxygen species contributions). We dispersed anaerobically grown CymA^GFP cells on a thin-film $Cu_2WS_4$ photoelectrode atop an ITO substrate electrode within a microfluidic photoelectrochemical cell (Fig. 2a(ii)). The GFP-tagging enabled fluorescence imaging of CymA

in the cell without the need for photoactivation, and CymA^GFP puncta are clearly visible expectedly (Supplementary Fig. 30). $Cu_2WS_4$, a stable photocathode material with a band gap excitable by 405 nm laser, has a conduction band edge sufficiently high to reduce protons in water and OM electron-transfer proteins (Supplementary Fig. 24)[33,37]. We adjusted the 405 nm laser illumination spot to be ~3 μm in diameter, slightly larger than the size of a single *S. oneidensis* cell (Fig. 4a lower and Supplementary Methods 1.9). When the illumination spot was placed on the $Cu_2WS_4$ film to excite charge carriers locally (Fig. 4a(iii), off-cell), a photocathodic current from proton reduction was clearly observable at a cathodic potential (i.e., −0.20 V vs. Ag/AgCl) when the laser was chopped on-and-off (Fig. 4b, blue)[38]. Strikingly, when the laser spot was placed on a nearby cell with CymA puncta (Fig. 4a(i)), an enhanced steady-state photocathodic current was clear and sustained over multiple light on-off cycles (Fig. 4b, red). This photocathodic current enhancement ($\Delta i$) quantifies a single cell's capability in electron uptake.

We measured $\Delta i$ for many cells containing CymA puncta. It averages at $-0.40 \pm 0.07$ (s.e.m.) nA, clearly negative (Fig. 4c, red) and consistent with bulk photoelectrochemistry measurements (Supplementary Fig. 31; see also Supplementary Notes 9 on discussions about the magnitude of $\Delta i$); it is also significantly larger than the off-cell control at $-0.20 \pm 0.06$ (s.e.m.) nA, measured as the difference between two locations on the $Cu_2WS_4$ film (Fig. 4c, blue). Some cells (<19%) have $\Delta i > 0$; we attribute this behavior to cell's displacement of water from $Cu_2WS_4$ surface, leading to suppressed proton reduction that dominates over these cells' photocathodic currents.

We also measured the $\Delta i$ on cells that had not formed CymA puncta (Fig. 4a(ii)); these cells had similar cellular CymA concentrations (Fig. 4d, top), consistent with that CymA's expression level is largely invariant (Fig. 1e, column 1-vs.-4). Remarkably, the $\Delta i$ of these cells does not differ significantly from the off-cell control (Fig. 4c, gray-vs.-blue). This indifferentiability between cells without puncta and off-cell control also confirms that the enhanced photocathodic current from cells with puncta is not due to the 405 nm laser increasing their photoconductivity in electron transfer networks (e.g., nanowire[39]). Note that our binary assignment of cells into with and without puncta has the complication of some mis-assignment of cells that have small but unrecognizable puncta, which may underscore the overlap between photocurrent distributions of the two types of cells in Fig. 4c.

Moreover, in cells either with or without puncta, their cellular CymA concentrations show no significant correlation with their $\Delta i$ (Fig. 4d) and just a marginal correlation with their photocurrent density (Supplementary Fig. 13 and Supplementary Note 10 for additional discussion between CymA concentration and EET function). Therefore, the presence of CymA alone at its physiological expression level is insufficient for EET; CymA must form puncta to enable EET.

We further compared the cellular concentrations of other EET partner proteins (i.e., STC, FccA, and OmcA) in anaerobically grown cells that had vs. had not formed CymA puncta; no discernible differences were observed (Supplementary Notes 11). Therefore, in transitioning into anaerobic growth with EET, the cell's EET capability is dominantly controlled by CymA punctum formation and not limited by the expression levels of these other EET proteins.

Previous studies showed that in cells pre-adapted to anaerobic growth, flavins can enhance EET by accelerating electron transfer through outer membrane cytochromes like MtrC/OmcA, suggesting these as rate-limiting steps[40–42]. In contrast, our data reveal that in transitioning from aerobic to anaerobic respiration, CymA puncta formation appears to be an upstream rate-limiting step that enables subsequent EET. Once CymA puncta are formed, the rate-limiting step can shift toward other downstream processes (e.g., those by Mtr/OmcA). Thus, our results complement existing literature and reveal a sequential regulation of rate-limiting steps in the overall EET pathway, which depends on the physiological state of the cell and the timing of protein spatial reorganization.

## CymA reorganization likely stems from biomolecular condensate formation

To further investigate the dynamic properties of CymA[PmC] in the cell, we examined its diffusivity using single-molecule tracking (Fig. 5a inset; Supplementary Methods 1.5). The displacement length distribution, corrected for the cell confinement effects through inverse transformation (Supplementary Notes 12)[43,44], allowed us to resolve two diffusion states of CymA[PmC] and their corresponding fractional populations (Fig. 5a). Under aerobic growth where CymA is not engaged in EET, the dominant fraction (~80%) of cellular CymA[PmC] exists in a state with a diffusion constant $D \sim 0.15 \, \mu m^2 \, s^{-1}$ (Fig. 5b, 1st column), typical of mobile membrane proteins[45] and which we attribute to CymA[PmC] located on the IM. The other minor fraction (~20%) has a much faster diffusion constant ($D = 5–7 \, \mu m^2 \, s^{-1}$), typical of free-diffusing proteins in the periplasm[46], and is attributable to those

CymA[PmC] in the periplasm; this attribution is consistent with CymA being known to have a minor periplasmic fraction, as it is easily removed from the IM due to its weak anchoring via a single helix[47]; this minor fraction could also comprise the <8% cleaved PmC tag seen in western blot (Supplementary Notes 1). To visualize the locations of CymA[PmC] more selectively in the dominant mobile state, we generated its super-resolution image using single-molecule localizations associated with smaller displacements. CymA[PmC] in this dominant state is distributed homogeneously over the cell envelope (Fig. 5c), consistent with its being IM-anchored and with the earlier diffraction-limited fluorescence images (Fig. 1b).

Surprisingly, under anaerobic growth where CymA actively participates in EET and forms punctate patterns in the cell, CymA[PmC] still exhibits a dominant mobile state with comparable fractional populations and slightly slower diffusion constants (~80%; $0.10–0.15 \, \mu m^2 \, s^{-1}$; Supplementary Table 4), regardless of the EET direction or the specific redox pairs (Fig. 5b, 2nd–5th columns). The corresponding super-resolution images show that CymA[PmC] in the dominant mobile state mostly resides in puncta, yet some are outside puncta (Fig. 5d-vs.-e). Analyzing the diffusion trajectories of CymA[PmC] within and outside puncta (e.g., Fig. 5e, red and green trajectories, respectively) reveals that the same dominant mobile state ($D \sim 0.10–0.15 \, \mu m^2 \, s^{-1}$) indeed exists in both regions (Supplementary Notes 13). Therefore, punctum formation does not substantially slow down CymA diffusion, and CymA maintains its mobility while its spatial distribution reorganizes into the punctate pattern when engaging in EET.

Furthermore, many single-molecule tracking trajectories of CymA[PmC] move in/out of puncta (Fig. 5e inset and Supplementary Fig. 33), indicating these puncta are membrane-less domains and exhibit dynamic exchange with the surrounding. We extracted the trajectories of single CymA[PmC] molecule's distances from the punctum centers (Fig. 5g, inset); thresholding such distance trajectories with the radii of the puncta allowed for extracting CymA[PmC]'s residence time, $\tau_{in}$, in the puncta. Analyzing the distribution of $\tau_{in}$ while also correcting for PmC's photobleaching/blinking kinetics gave CymA's rate constant ($k_e$) of escaping puncta at ~3.1 $s^{-1}$ (Fig. 5g). We further deduced the approximate rate constant ($k_c$) of CymA in moving into puncta, at ~0.6 $s^{-1}$, assuming a quasi-equilibrium of CymA molecules being in and out of puncta (Supplementary Notes 14). Additional fluorescence recovery after photobleaching measurements further confirmed the dynamic exchange of CymA (Supplementary Fig. 32). The dynamic exchange in/out of puncta and the mobility within the puncta both rule against the possibilities of CymA forming highly ordered state or misfolded aggregates in the puncta. Consistently, bright-field transmission images of the punctum-containing cells do not show bright-spot-like features of inclusion bodies (Fig. 5f-vs.-e).

As we showed above that neither of the direct or indirect electron-transfer partners is the cause of CymA punctum formation during EET, we postulated that the small molecule menaquinone may play a role, given its function as an IM redox mediator that *directly* exchanges electrons with CymA[12,48] (Fig. 1a). It is important to note that self-interactions among CymA proteins is likely not a cause either, as CymA diffusivity does not slow down during punctum formation (Fig. 5b and Supplementary Notes 13). It is known that quinones and polar lipids can form raft-like microdomains relevant for electron transport[49]. Unfortunately, there is no accessible way to map the spatial distribution of menaquinone—a small, non-genetically editable molecule—on the IM in living bacteria; deleting menaquinone synthase is also lethal for anaerobic respiration (Supplementary Fig. 34). However, it is known that menaquinone level is upregulated in *S. oneidensis* under anaerobic growth[12]. We therefore assayed menaquinone production in cells over varying periods of anaerobic growth with active EET. This bulk cell assay indeed revealed a menaquinone level increase over time, which, notably, occurs with different timescales under uptake-vs-outflow EET conditions, at ~2.0 and ~6 h, respectively

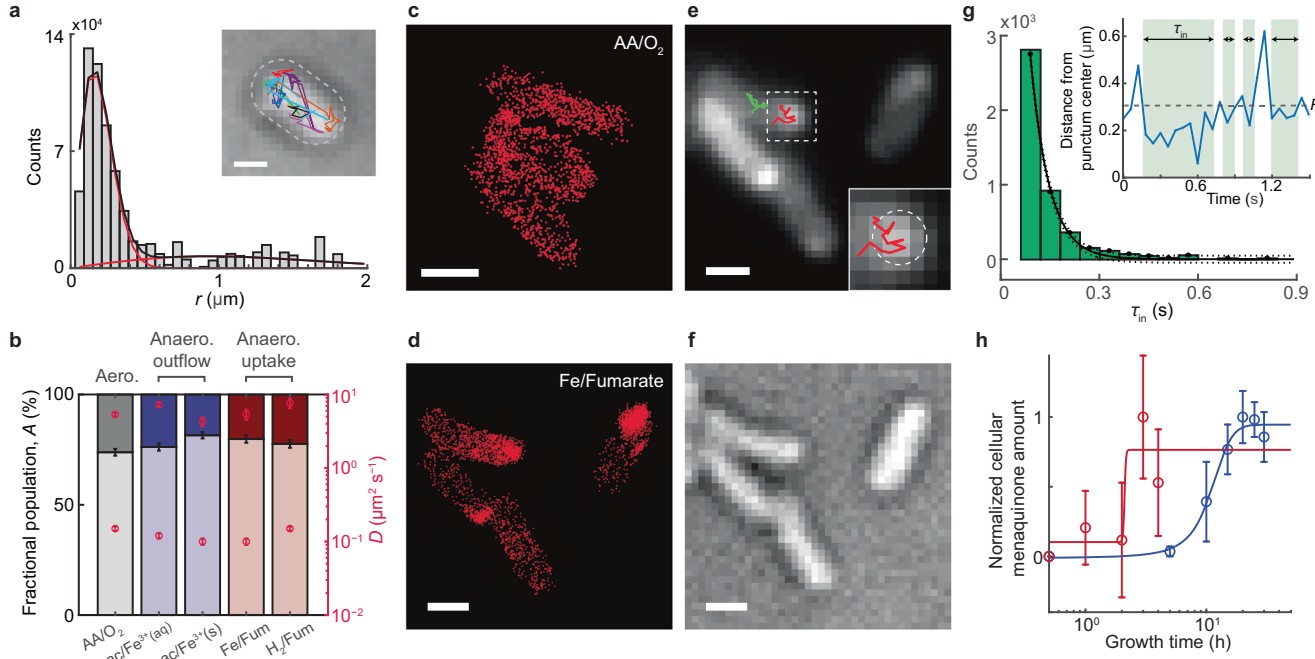

**Fig. 5 | CymA puncta during EET are likely biomolecular condensates. a** Cell-confinement-effect−deconvoluted displacement length $r$ distribution of CymA$^{PmC}$ from single-molecule tracking in 269 cells grown aerobically. Lines: overall fit (black) and the two resolved Brownian diffusion states (red). Inset: representative single-molecule tracking trajectories (colored lines) overlaid on the bright-field transmission image of a cell; white dashed line: cell boundary; scale bar: 1 μm. **b** Fractional populations of the two diffusion states (bars, left axis) and their corresponding diffusion constants (pink open circles, right axis) of CymA$^{PmC}$ in cells grown aerobically (column 1) and anaerobically with outflow (columns 2–3) and uptake (columns 4–5) EET. Error bars: 95% confidence interval from the fitting. **c** Reconstructed super-resolution image of CymA$^{PmC}$ in the dominant mobile diffusion state in cells grown aerobically (Supplementary Notes 13.1). **d** Same as (**c**) but for cells grown anaerobically with metal iron/fumarate as the electron donor/acceptor for uptake EET. **e, f** Wide-field fluorescence image of ensemble-photoactivated CymA$^{PmC}$ (**e**) and bright-field transmission image (**f**) of the cells in

(**d**). Red/green lines in **e**: two exemplary single-molecule tracking trajectories, one primarily in a condensate (red) and one outside a condensate (green); inset in **e**: zoom-in of the dashed box; dashed circle: condensate outline. Scale bars in (**c**–**f**): 0.5 μm. **g** Distribution of CymA$^{PmC}$'s residence time $\tau_{in}$ in condensates. Solid line: single exponential fit with 95% confidence bounds (dotted lines), with the decay rate constant $k_{eff} = k_{bl}(t_{int}/t_{tl}) + k_e$, where $k_{bl}$ is the photobleaching/blinking rate constant of PmC, $t_{int}$ is the laser pulse duration, $t_{tl}$ is the image time-lapse, and $k_e$ is the escaping rate constant (Supplementary Notes 14.1). Inset: Exemplary trajectory of a single CymA$^{PmC}$ molecule's distance from the center of a condensate. Residence time ($\tau_{in}$) of the molecule inside the condensate is thresholded by $R$ (dashed line), the fitted radius of the condensate. **h** Normalized cellular menaquinone production vs. time under anaerobic growth with uptake (metal iron/fumarate, red circles) or outflow (lactate/ferric citrate, blue circles) EET. Data are presented as mean ± s.d. from 3 biological replicates for each. Lines: sigmoidal fits. Source data are provided in a Source Data file.

(Fig. 5h, red vs. blue). Remarkably, such two timescales largely match the respective timescales (~2.7 and ~7 h) of CymA punctum formation from population analysis (Fig. 1g). These temporal coincidences strongly suggest the involvement of menaquinone in rendering CymA punctum formation (Supplementary Fig. 35).

On the basis of: (1) the spatially condensed nature, (2) reversible formation comprising temporal emergence and dissipation, (3) molecular mobility, and (4) dynamic exchange and coexistence with outside CymA populations, which are all hallmarks of biomolecular condensates[50], we propose that the CymA puncta are part of membrane condensates (a.k.a., liquid-liquid phase separated domains), which are enriched with the anaerobic electron carrier menaquinone, whose upregulation coincides with CymA punctum formation. Although CymA puncta tend to locate closer to cell poles (Supplementary Fig. 20), their formation must not directly result from cell division, as cell division is much slower (6–8 h[51]) under our anaerobic EET conditions, while these puncta can form as fast as in less than 1 h (Fig. 3e, f). Moreover, cells with puncta are generally not actively dividing (Supplementary Fig. 21, Supplementary Fig. 25 and Supplementary Fig. 30). Nor should nucleoid occlusion be the underlying cause, as CymA is a membrane protein. The polar localization of CymA condensates could be from the lipid heterogeneity of these menaquinone-enriched IM domains, which might contain a certain type of lipid (e.g., cardiolipin) that could give rise to preferences for the cell poles[52].

We further confirmed the reversibility of CymA punctum formation in response to external condition changes, another hallmark of biomolecular condensates[50,53]: when anaerobically grown cells that contain CymA puncta were re-introduced to aerobic growth, the CymA puncta dissipated, and CymA became uniformly distributed across the cell envelope (Supplementary Fig. 36). The fact that the diffusion coefficient of the dominant mobile state of CymA becomes just slightly slower upon forming condensates is also in agreement with literature: the apparent diffusion coefficient of HslU (ATP-dependent protease) remained similar from $0.12 \pm 0.07 \, \mu m^2/s$ (early stage) to $0.10 \pm 0.03 \, \mu m^2/s$ (late stage) upon forming condensate[54]; PopTag$^{SL}$ and PopTag$^{LL}$, (polar organizing protein Z) and McdB (maintenance of carboxysome distribution protein B) displayed a modest reduction, varying by proteins' properties[50].

## Discussion

We have shown that in the electroactive bacterium *S. oneidensis*, the IM electron-transfer hub protein CymA undergoes spatial reorganization to form localized, dynamic puncta, which colocalize with its direct electron-transfer partners STC and FccA in the periplasm, to enable EET (see overall mechanistic scheme in Supplementary Fig. 35). These CymA puncta are likely biomolecular condensates associated with IM domains enriched with the anaerobic electron carrier menaquinone, which should facilitate the electron exchange between CymA and menaquinone. The dynamic mobility of CymA within these

condensates, as well as the colocalization and dynamic interactions with STC/FccA, may also enable STC/FccA to sample multiple CymA molecules to find those with suitable redox states for effective electron exchange before STC/FccA molecules diffuse toward EET proteins on the OM (Supplementary Fig. 37).

It is also worth noting that investigating the function of biomolecular condensates poses challenges in designing control experiments that only modify the spatial distribution of a protein without interrupting its other functions[55]. Our correlated single-cell-level photoelectrochemistry and fluorescence imaging compares cells with and without condensates (Fig. 4), thus directly informing on the functional roles of condensate formation in the physiological environment.

Another critical characteristic of biomolecular condensates is their concentration dependency, in which they typically form when a saturation concentration is reached. Here, the cellular concentration of CymA remains almost invariant regardless of EET engagement and punctum formation (Fig. 1e). We postulate that this is because the CymA condensates are membrane-based condensates, in which menaquinone-enriched raft-like domains perhaps act as scaffolds with CymA serving as a client, especially because lipid membrane domains often underlie biomolecular condensate formation in cells[56,57]. Furthermore, we did not observe CymA puncta fusion in cells. It is possible that condensate growth may favor over fast nucleation and fusion or that fusion occurs only in early stages and is not observable with mature condensate droplets. Further in vitro studies are needed to explore this aspect.

Increasing evidence also indicates that biomolecular condensates play key roles in subcellular spatial organizations: they are prevalent in membranes and cytoplasm from bacterial to eukaryotic cells[53,58,59]. Proteins within condensates often engage in functions that require interactions with partners in or outside condensates (such as for electron transfer studied here), nucleotide biosynthesis, reaction activation, signal transduction, cytoskeletal structure nucleation, localization, etc.[58,60]. Therefore, it is not unreasonable to postulate that condensate formation for enabling function might be broadly applicable—for example, nitrate reductase in *E. coli* exhibits subcellular localizations, enhancing nitrate reduction activity[61]; various respiratory proteins localize in response to nutrient starvation in aerobic conditions[54], implying a functional role in cellular fitness. Cytochrome *c* can be recruited along with its nuclear partners into a biomolecular condensate, facilitating chromatin remodeling and DNA damage response[62]. Quantifying the spatiotemporal dynamics of these proteins in cells in correlation with their cellular functions of EET at the single-cell level suggests that the relevance of condensates may extend beyond previously recognized contexts, potentially influencing a broader range of biological functions.

## Methods
### Statistics and reproducibility
In Fig. 1b-d, wide-field fluorescence images of ensemble-photoactivated CymA[PmC] in *S. oneidensis* grown under aerobic (b) or anaerobic (c,d) conditions with different electron donor/terminal electron acceptor pairs for outflow (c, lactate/ferric citrate) or uptake (d, metal iron/fumarate) EET. These are representative images collected from at least 12 movies with hundreds of cells; $n = 269, 481, 272$. In Fig. 1c, cellular concentrations of CymA[PmC] grown aerobically (column 1, $n = 269$) or anaerobically with different electron donor/acceptor pairs for outflow (columns 2-3; $n = 481$ and 121) or uptake (columns 4−5; $n = 272$ and 167) EET. Data collected from 2 to 3 biological replicates. In Fig. 1f, fraction of cells showing CymA[PmC] puncta after 2-day aerobic or anaerobic growth with outflow/uptake EET. (Number of images: 38, 14, 10, 5, and 8; total number of cells: 821, 420, 135, 151, and 140; for columns 1−5, respectively). Means and error bars (s.d.) are from multiple images from 2 to 3 biological replicates, with the

calculations weighed by the cell counts. In Fig. 1g, fraction of cells exhibiting CymA[PmC] puncta vs. time under anaerobic growth with outflow (blue circles, lactate/ferric citrate as the redox pair) and uptake (red circles, metal iron/fumarate as the redox pair) EET. Different time points are from different sets of cells. (Outflow: number of images: 5, 8, 10, 11, 10, 10, and 14; total number of cells: 100, 295, 165, 286, 268, 213, and 420 for a total of 7 time points in increasing order; uptake: number of images: 10, 10, 10, 10, 9, 10, 12, 10, 7, and 5; total number of cells: 195, 166, 165, 64, 131, 60, 131, 60, 78, 224, 60, and 151 for a total of 10 time points in increasing order). Lines: fits with a sigmoidal function $f(t) = a + \frac{b}{1+\exp(-(t-c)/d)}$. Transition time is defined as four times the exponential time constant $d$, straddling the inflection point. Lag time is defined as the period from $t = 0$ to the start of the transition time (Supplementary Notes 4). Means and error bars (s.e.m.) are from multiple images from 1 to 3 biological replicates, with the calculations weighed by the cell counts. In Fig. 1h, means and error bars (s.d.) are from 3 biological replicates.

In Fig. 5a, the cell-confinement-effect−deconvoluted displacement length $r$ distribution of CymA[PmC] from single-molecule tracking in 269 cells grown aerobically. In Fig. 5b, fractional populations of the two diffusion states (bars, left axis) and their corresponding diffusion constants of CymA[PmC] in cells grown aerobically (column 1, $n = 269$) and anaerobically with outflow (columns 2−3, $n = 481$ and 121) and uptake (columns 4-5, $n = 272$ and 167) EET. Error bars: 95% confidence interval from the fitting. In Fig. 5e, f, a wide-field fluorescence image of ensemble-photoactivated CymA[PmC] (e) and bright-field transmission image (f) of the cells in (d). These are representative images collected from 12 movies with hundreds of cells ($n = 272$). In Fig. 5g, 4563 residence times were collected from 346 cells. In Fig. 5h, data are presented as mean ± s.d. from 3 biological replicates for each. Extracted lag times for outflow and uptake are $2.0 \pm 0.1$ and $6 \pm 1$ h, respectively (errors: s.e.).

### Bacterial strains and growth conditions
*Shewanella oneidensis* MR-1 and *E. coli* WM3064 (developed by William Metcalf from UIUC, unpublished) were obtained from Buz Barstow at Cornell University. For aerobic growth, *S. oneidensis* was grown in LB at 30 °C for 20 h, then diluted to minimal medium (MM) supplemented with amino acids and vitamins, followed by further growth to $OD_{600}$ of -0.4. For anaerobic growth condition, *S. oneidensis* was initially grown in LB at 30 °C for 20 h aerobically, then diluted to MM supplemented with an electron donor and acceptor, followed by purging with $N_2$ gas, and subsequently grown at 30 °C for 2 days. More details are in Supplementary Methods 1.1.

### Strain construction
All genetic engineering was done by biparental conjugation method[38,63]. Briefly, for fluorescent protein tagging, the conjugation plasmid was constructed by Gibson assembly of HR1, HR2 and the fluorescent protein gene into a pRE118 vector. HR1 and HR2 are the 1000-bp homology regions before and after the stop codon of the target gene, respectively. For gene knockout, only HR1 and HR2 were cloned into the conjugation vector, where HR1 and HR2 are regions before and after the entire target gene. The conjugation plasmid was transformed into the donor *Escherichia coli* strain WM3064, which was then conjugated with *S. oneidensis*. Colonies with correct double crossover were screened by PCR and sequence-confirmed. More details are in Supplementary Methods 1.2

### Wide-field fluorescence imaging for single-molecule tracking and protein quantification
For fluorescence imaging, concentrated cells were immobilized between a glass coverslip and an agarose pad. To prevent gel drying and medium evaporation, the coverslip edges were sealed with epoxy. Imaging was conducted with a 60× oil immersion objective (Olympus

PlanApo N 60× oil 1.45 TIRFM UIS 2) on an Olympus IX71 inverted microscope equipped with transmission optics. An EMCCD camera (Andor Technology, DU-897E-CSO-#BV, pixel size $16 \times 16\ \mu m^2$) served as the detector. Coaxially aligned continuous wave lasers (405 nm, CrystaLaser, DL405-100; 488 nm, Coherent, Sapphire 488-100 CW CDRH; 561 nm, Coherent, Sapphire 561-100 CW CDRH) passed through dichroic filters (Chroma, T510lpxru and T425lpxr) and a broadband quarter waveplate to be changed from being linear polarized to circular polarizization. The lasers, expanded four times by an achromatic lens pair and focused (40 cm lens) at the back focal plane of the objective, were then reflected toward the objective by a three-band dichroic filter (Chroma, ZT405/488/561rpc) inside the turret cube. In the emission detection path, a bandpass filter (Semrock, FF01-617/73) was used for detecting the red fluorescence from photoactivated PAmCherry1, while a bandpass filter (Chroma, ET525/50 m) was used for detecting the green fluorescence from sfGFP or SynaptoGreen. For single-molecule tracking, a 405 nm laser ($2-5\ W/cm^2$) was used to photoactivate a PAmCherry1 molecule, then tracked with 561 nm laser ($\sim 3\ kW/cm^2$), with 4 ms exposure time and a time-lapse of 60 ms in stroboscopic imaging mode. The fluorescence was collected by an EMCCD camera with 250 EM gain, synchronized with the 561 nm laser pulses. This imaging procedure was iterated for 500 cycles. We used custom-written MATLAB code to analyze the tracking movies[64,65]. For protein quantification, 405 nm laser ($\sim 5\ W/cm^2$) illuminated cells for 1 min to photoactivate all fluorescent proteins, followed by 561 nm laser illumination for 2000 frames. This 405-illumination and 561-excitation sequence was repeated twice to ensure all PAmCherry1 molecules had been photoactivated and imaged. Copy number was calculated as the whole-cell fluorescence intensity divided by the single fluorescent protein intensity. Single-molecule intensity was directly obtained from the single-molecule tracking process. The procedure is conducted similarly as we reported previously[38,44,64,66]. More details are in Supplementary Methods 1.5.

### Single-cell level electrochemical fluorescence imaging

For electrochemical fluorescence imaging of bacteria, cells were deposited on ITO-coated glass to have direct contact with the ITO electrode and sandwiched with an agarose pad for immobilization. The detailed electrochemical microfluidic cell construction is shown in Supplementary Fig. 1c. After the assembled microfluidic cell was completely sealed with epoxy, the minimum medium purged with $N_2$ was introduced to have an anaerobic condition, then, an electrochemical potential was applied. The ITO electrode served as the working electrode, the electrochemical potential of which was controlled by a potentiostat (CH Instrument, CHI1200a). A platinum wire and an Ag/AgCl electrode (BASi, MW-2030) were used as the counter and reference electrode, respectively, and were placed in a liquid chamber about ~3 cm downstream. Imaging was conducted with the same microscope setup used for wide-field fluorescence imaging except for a 60× water immersion objective (Olympus, UPlanSApo 60× NA 1.20 water UIS 2). Cells were illuminated with the 405 nm laser ($\sim 5\ W/cm^2$) for 2 min to photoactivate all fluorescent protein molecules, followed by 561 nm laser ($\sim 1.5\ kW/cm^2$) illumination with 15 ms exposure time with 15–180 min interval. More details are in Supplementary Methods 1.6.

### Integrated photoelectrochemical current measurement and fluorescence imaging

Photoelectrochemical current measurement and fluorescence imaging at the single-cell level utilized the same optical configuration as electrochemical fluorescence imaging (Supplementary Fig. 1c). Coaxially aligned continuous wave lasers (405 nm, Coherent, OBIS 1265575; 488 nm, CrystaLser DL488-050) passed through dichroic filters (Thorlab, DMLP425) and a broadband quarter-waveplate to convert linear polarized light to be circular polarized. A suitable combination

of a flip lens (Thorlabs, N-BK7 Plano-Convex Lens) mounted on an XYZ translation stage (Thorlabs, PT3-1, 1/4-20 Taps) placed near the back port of the microscope enabled switching between wide-field fluorescence imaging and the focused laser illumination for single-cell photocurrent measurements. The lasers were then directed toward the objective by a dichroic filter (Chroma, ZT405rdc and ZT375/488/532/635rpc) inside the turret cube. In the emission path, a bandpass filter (Semrock, FF01-531/40) was employed for detecting green fluorescence from sfGFP. Cells were deposited on a $Cu_2WS_4$ thin-film on top of an ITO electrode, similarly as we reported for measuring other bacterial cells[38]. Cells were illuminated with the 488 nm laser ($\sim 5\ W/cm^2$) to excite and acquire sfGFP fluorescence images, then a 405-nm laser, exciting the charge carriers in $Cu_2WS_4$, underwent enlargement via a beam expander, was directed to cell. Laser spot was adjusted to cover an entire single cell sufficiently ($\sim 3\ \mu m$ diameter). This laser spot could be positioned either on top of a cell (on-cell location) or near the cell (off-cell location) by moving the sample stage. Upon placing the laser spot at the desired location, chronoamperometric measurements at $-0.2\ V$ (vs. Ag/AgCl) were conducted during light-chopping, typically involving 15 s of on-period followed by 10 s of off-period for three cycles. To quantify the photocurrent, the currents for the 2–3 s durations on either side of the time of turning illumination on, ($t_{on} - 3\ s$, $t_{on} - 1\ s$) and ($t_{on} + 1\ s$, $t_{on} + 4\ s$), were fitted with two different linear functions. The difference in these two linear functions at $t_{on}$ was taken as the photocurrent (i.e., $i_{ph}$) for this light on/off cycle. More details are in Supplementary Methods 1.9.

### Reporting summary

Further information on research design is available in the Nature Portfolio Reporting Summary linked to this article.

## Data availability

All data are available in the main text or the Supplementary Information. The source data underlying Figs. 1, 2, 4, 5, and Supplementary Figs. 2, 3, 4, 7, 9, 10, 13, 14, 15, 16, 17, 18, 19, 20, 22, 23, 26, 27, 28, 29, 31, 32, 34, and 36 are provided in a Source Data file. Raw imaging data supporting the findings of this study are available upon request. Source data are provided with this paper.

## Code availability

MATLAB codes for data analysis are included in Supplementary Code 1.

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

## Acknowledgements

This research is supported by NIH Grant GM154669; early studies were supported by NIH Grant GM109993. We thank previous lab members, Drs. Xianwen Mao and Wenjie Li, for discussion on photoelectrochemical measurement and electrochemical setup; and the Cornell Institute of Biotechnology's Imaging Facility with NYSTEM (CO29155) and NIH (S10OD018516) funding for the shared Zeiss LSM880 confocal/multi-photon microscope.

## Author contributions

Y.P. designed and performed experiments, constructed strains, wrote MATLAB codes, analyzed data; T.Y. contributed to strain construction and live-cell imaging experiments; Z.Z. contributed to single-cell photoelectrochemical experiment; B.F. contributed to discussion and helped with code writing; M.Y. contributed to bulk photoelectrochemical experiment; F.S. and B.B. contributed to strain construction; P.C. directed the research. Y.P. and P.C. wrote the manuscript.

## Competing interests

The authors declare no competing interests.
