## [Transparent Peer Review File · Nature Communications]

Spatiotemporal organization of membrane protein controls bacterial extracellular electron transfer

Corresponding Author: Professor Peng Chen

Version 0:

Reviewer comments:

Reviewer #1

(Remarks to the Author)

In this paper, the authors successfully combined electrochemical measurements with single-cell or single-molecule fluorescence microscopy. This innovative approach allows for the visualization of the aggregate formation of CymA, suggesting its link with electron flow. With the observation of the dynamics of CymA in individual cells and meticulous analysis, the study also demonstrates the heterogeneity of aggregate formation among different cells. The authors developed a model that colocalizes CymA aggregation with FccA/STC aggregation at the cell poles, facilitating efficient electron flow from the inside of the cell to the exterior. This model is fascinating and is particularly appealing in the context of the respiratory electron transport chain as well as the bacterial extracellular electron transfer (EET) pathway. While the biomolecules involved in the electron transport chain have been identified for some time, the dynamics and spatial organization of respiratory proteins have not been thoroughly explored due to technological limitations. This study highlights the significance of protein aggregate formation (possibly liquid-liquid phase separated domains) in the respiratory process. However, the relationship between aggregate formation and the proximity of CymA to FccA/STC is not sufficiently supported by the data presented, and this issue should be addressed prior to publication. Corrections and suggestions are provided below:

Major concerns:

1. In this paper, the formation of CymA, FccA, and STC aggregates at the cell pole was demonstrated, and a correlation between aggregate formation and extracellular electron transfer (EET) kinetics was suggested. The authors state that “the spatial colocalization of CymA’s direct electron-transfer partners, STC and FccA, which leads to geometric proximity and are thus expected to facilitate EET” (page 9). However, the connection between aggregate formation and the proximity of the proteins is not clear. This is because the aggregates consist of the same protein and do not involve mixing (no condensates containing both CymA and FccA/STC were observed). To clarify this relationship, I recommend quantitatively demonstrating the impact of aggregate formation on the distance between CymA and the nearest FccA/STC. This can be achieved by calculating the distances based on single-molecule tracking under four conditions: CymA (+, - condensate) versus FccA/STC (+, - condensate). Analyzing single-molecule tracking data would enable this quantification effectively.
2. The EET process requires the shuttle of FccA/STC from the inner membrane to the outer membrane as mentioned on page 15. Aggregate formation of FccA/STC could hinder efficient diffusion in the periplasmic space due to the increased protein size and the reduced number of FccA/STC available for collision. Please discuss the potential disadvantages of aggregate formation on the efficient EET in the discussion part.
3. In this study, the author employed photocurrent measurement for assessing single-cell extracellular electron transfer (EET). While the idea and experimental setup are commendable, validating this system poses challenges. Although the authors provided detailed explanations of single-cell photocurrent measurement in supplementary notes 7 and 8, it remains difficult to conclude that the formation of CymA aggregates leads to enhanced EET kinetics. The photocurrent measurement using Cu₂WS₄ generates reactive oxygen species (ROS), and the presence of reactants may facilitate photocurrent generation. Additionally, biomolecular condensates can alter their redox properties, including their reactivity with ROS (Y. Dai et al, Chem, 2023, 9 (6), 1594). Thus, it is plausible that the formation of CymA aggregates simply enhances ROS consumption and photocurrent generation, rather than improving EET kinetics or the catalytic conversion of fumarate to succinate. To establish a clearer connection between CymA aggregation and EET, I recommend measuring ROS levels using appropriate indicators during the photocurrent measurement.

4. On pages 5-6, the authors mention that "We imaged the photoactivated CymAPmC in each cell over 6 h, during which there was no substantial new synthesis of CymAPmC in the cell (Supplementary Notes 3). After 6 h, CymAPmC puncta were observed in many cells (Fig. 2b, right, and Supplementary Fig. 20a, b)." The supplementary figure 5b in supplementary notes 3 shows CymA concentration in the cells across different images captured at 25-30 min intervals. They showed 8 data in anaerobically grown cells, thus, I think that no synthesis of CymAPmC is guaranteed only for 4 h at maximum.

5. Although cell division takes 6-8 hours under anaerobic conditions, as noted on page 13, some cells observed at the maximum of 4 hours in supplementary figure 5b should have undergone division (if doubling time is 8 hours, about half of the cells should divide in 4 hours). This is because not all cells synchronize their division timing. Given that cell division would dilute the matured CymAPmC concentration in each cell, it is puzzling why the concentration of matured CymAPmC remains unchanged after 4 hours of cultivation. A rational explanation for this observation is needed.

6. The authors demonstrate that the reduction of Fe(III) requires a lag time, which corresponds with the formation of CymA aggregates (see Figure 1). Along with their claim, it is suggested that the kinetics of extracellular electron transfer (EET) may be limited by the formation of these aggregates. However, accumulating evidence indicates that the acceleration of electron transfer in c-type cytochromes located on the outer membrane, such as MtrC and OmcA, is enhanced by flavins and redox molecules, leading to an increase in EET kinetics by several-fold. This suggests that MtrC/OmcA may be the rate-limiting step in EET (E. Marsili et al., PNAS, 2008, 105(10), 3968; H. Canstein et al., Appl. Environ. Microbiol., 2008, 74(3)). Notably, this acceleration occurs rapidly (within several minutes) during the electron uptake process, which means it would not induce CymA aggregation and would primarily influence the electron flow across the outer membrane (A. Okamoto et al., Angew. Chem. Int. Ed., 2014, 53(41), 10988; Y. Tokunou et al., J. Phys. Chem. C, 2016, 120(29), 16168). This presents a conflict between the current data and traditional studies on this topic, warranting further explanation in the discussion part.

7. Although the authors suggested that menaquinone reduction drives CymA aggregate formation, the mechanisms underlying the FccA/STC aggregate formation are totally unclear. Please discuss possible models for this phenomenon.

Minor concerns:

1. The combination of iron metal and fumarate, used as an electron donor and acceptor (mentioned on page 3), is quite rare, I guess. Please include a reference that discusses this combination.

2. While photocurrent data is presented in Figure 4, the chronoamperometric data related to Figure 2 are missing. I recommend including this data in the Supplementary Information.

3. In this paper, Ag/AgCl is used as the standard potential, but the potential of Ag/AgCl changes according to the Cl⁻ concentration. Please state the Cl⁻ concentration somewhere (e.g. Ag/AgCl KCl sat., Ag/AgCl 3M KCl).

(Remarks on code availability)

Reviewer #2

(Remarks to the Author)

The manuscript reports on the spatio-temporal coordination of protein complexes involved in extracellular electron transfer (EET) in the electroactive bacterium *Shewanella oneidensis*. Perk et al. integrate several experimental approaches to investigate the functional significance of the spatial organization of the hub protein CymA, particularly in relation to its direct electron-transfer partners. Ultimately, the authors attribute this reorganization to the formation of a biomolecular condensate. Their use of a previously developed imaging platform to quantify the functional impact of spatial compartmentalization in electroactive bacteria is promising, as it seeks to directly link protein localization to functional outcomes. However, the manuscript emphasizes protein aggregation via biomolecular condensate formation as the central finding, yet the experimental data to support this conclusion are fragmented and partial. Several aspects of the study are reported at a superficial level, with key statistical details missing.

- Overexpression. What role does overexpression play in the observed spatial compartmentalization? The authors note that "overexpressing CymA shows higher EET activities" (Supplementary Fig. 25), and Figure 2g shows a correlation between CymA copy number and lag time. These observations suggest that expression levels may significantly influence both spatial organization and EET function. A more thorough discussion of this potential relationship would strengthen the interpretation of the results.

- Reversibility. This critical aspect is only briefly mentioned and not quantitatively addressed, despite its importance for identifying biomolecular condensates, as noted in the literature and by the authors themselves (page 13, lines 7–33). Specifically, the lack of quantification in Supplementary Figure 29 limits a full understanding of the reversibility process and its role in the observed phenomena. Moreover, the distribution shown in panels g–i appears more homogeneous compared to panels a–c in the same figure, raising further questions: how efficient is the reversibility? Can it be visualized in the same sample over time to track the process dynamically? Additionally, what is the viability of the cells after the assay? On page 3, line 25, how does reversibility impact the quantification of the number of puncta? In the electrochemical

manipulation assay, the system remains under anaerobic conditions, so what is the nature of the 10% reversibility mentioned?

- Dynamic exchange. In Supplementary Note 11, no clear difference in the topological distribution of the two diffusive populations is visible in the images presented in Supplementary Figure 12. The population with higher diffusivity appears less dense, likely due to its smaller fraction, but there is no apparent spatial separation that would support the hypothesis that “raft-like lipid domains, formed with menaquinone and the membrane, are involved in the formation of CymA puncta.” This observation alone does not sufficiently support the proposed model. Additionally, in Supplementary Note 11.2, the percentages reported lack associated error estimates, making it difficult to assess the robustness of the data. How large is the statistical sample used in this analysis? Could the periplasmic fraction be quantified more rigorously? Would super-resolution techniques such as SMLM provide a better 3D characterization of CymA distribution? While the authors attempt this in Figure 5a–c, it is unclear how 3D diffusion affects the thresholding applied to classify step sizes. How robust is the classification method used to sort diffusion steps? Simulations of various diffusion modes within a realistic bacterial geometry could help evaluate and validate the applied analysis pipeline. The discussion of diffusion coefficients is also confusing. In one instance (page 13, lines 25–26), the diffusion is described as “slightly slower,” whereas in another (page 12, line 12), it is “similar/slightly slower.” This inconsistency, along with the lack of statistical replicates (how many independent experiments were performed?), makes it difficult to assess whether a meaningful change in diffusivity is present. If, the diffusion coefficient is confirmed to remain unchanged, the cited literature only partially supports the authors’ conclusion that this is indicative of a biomolecular condensate. In the referenced study (page 13, lines 26–29), the lack of change in diffusion was observed only after correcting for centroid tracking of the aggregate. Has a similar correction been applied here? Overall, to more convincingly establish the biomolecular condensate identity of the CymA assemblies, a photobleaching assay (e.g., FRAP) should be incorporated into the validation strategy, in line with established approaches in the field.

Other major comments

- Pag. 1-line 27-28, pag 3 line 5-6 and pag. 15 line 22-34. The authors close both the abstract and the conclusion paragraph with a digression on the relevance of biomolecular condensate that sounds vague and difficult to link to the directly presented work. What do the authors specifically mean when they say that the observation could have a “broad relevance to the function of biomolecular condensates in biology”. A clearer description of why the condensate is important for the specific system would give a better closure to their finding and more easily identify the context and scope of the work.
- Pag. 3 line 23. The authors conclude that the aggregation is diffraction-limited, and at this level of spatial precision, all the conditions result in the same aggregation. Since they have the capability to perform single molecule localization microscopy in their integrated platform and with the fluorescent protein tag used, would super resolution provide a finer description of this association?
- Pag. 6 line 27. How does photobleaching affect the assay reported in Figure 2? Could it introduce a bias, particularly over longer time scales? If the emergence of a skewed distribution is being used as an indicator of aggregation, how robust is the temporal characterization—specifically the sigmoidal fit used by the authors? From the examples provided, it appears that the dynamics of aggregate formation may occur on a faster timescale than that captured by the current timelapse acquisition. This could potentially explain the lack of correlation observed for the transition time. This interpretation is further supported by Supplementary Figure 21b, where the transition times are heavily skewed toward zero. Would a faster imaging rate allow for a more accurate temporal description of the aggregation process? Improved time resolution might clarify the onset and progression of puncta formation and enhance the reliability of the fitted kinetic models.
- Pag. 8 line 20-27. The author rightfully dedicated care to the validation of the CymAPmC distribution, but no images of the CymAGFP are present. Especially in figure 3 it would be relevant to see both channels for all the presented condition and not only for the pair a-b and e-f.
- Pag. 10 line 29 and Supplementary Note 9 Fig 10. The concentration of the protein is often reported as an important parameter in the assay. For the partner proteins, there is more protein concentration in anaerobic conditions independently of the presence or absence of the condensate, and at the same time, the condensate correlates with higher photocurrent. How does CymA copy number vary in this picture?
- Instead of expressions like “first-of-its-kind” or “first”, it would be more valuable to directly address the status of the field until the presented work and how the approach introduces an important new observation.
- All the paper is based on the single-cell quantification of protein concentration, although based on previous literature, I would appreciate a clear explanation of the method, even just in the supplementary, explaining the limitations of such quantification. Furthermore, is this the first time the author used PAmCherry for this assay? How does the different photophysics of the protein enter the quantification? How does bleaching enter into this estimation? For example, if the diffuse population falls below the SNR level faster than the aggregate.

Minor comments

- Pag. 2 line 29-31. The sentence misses the verb, and the meaning is not clear.
- Supplementary fig 25 is a repetition of figure 4d.

(Remarks on code availability)

The code is provided and commented in its components, but the inclusion of example datasets would be essential for accessibility and practical use.

Version 1:

Reviewer comments:

Reviewer #1

(Remarks to the Author)

The authors have provided logical and appropriate responses to the concerns I raised in the previous round of review. While some of the experimental issues could not be addressed with additional data, the authors have thoroughly explained the technical limitations and convincingly argued that the lack of these experiments does not undermine the main conclusions. I recommend acceptance of the manuscript after minor revisions.

Comment on Reply 1–2:

In response to my previous concern that “Aggregate formation of FccA/STC could hinder efficient diffusion in the periplasmic space due to the increased protein size and the reduced number of FccA/STC available for collision. Please discuss the potential disadvantages of aggregate formation on the efficient EET in the discussion part,” the authors replied that “This dynamic association/dissociation of STC/FccA with CymA, coupled with the rapid diffusion of free periplasmic components, should allow efficient ET between CymA on the inner membrane and the Mtr complex on the outer membrane, mediated by FccA and STC.”

I appreciate the authors' effort to clarify that the FccA/STC puncta do not represent static aggregates but rather dynamic assemblies undergoing continuous association and dissociation with CymA. This addresses my initial assumption and adds valuable context to the functional interpretation.

However, I remain concerned about this explanation. The manuscript's interpretation of FccA/STC colocalization with CymA is understandable if one assumes there are no intermolecular interactions among FccA or STC themselves. However, there is no clear experimental evidence supporting the absence of such interactions during puncta formation. It would be rather reasonable to assume that FccA/STC has some interaction, as FccA/STC is condensed, similar to CymA, which potentially exchanges electrons among CymAs, as the author mentioned. Therefore, it remains reasonable to consider that punctum formation could have a negative impact on EET efficiency.

Moreover, in Figure S34, the authors additionally stated that “(v) STC and FccA can subsequently dissociate from the condensate and diffuse toward the outer-membrane to exchange electrons with the MtrABC/OmcA complex.” If it were shown that reduced FccA/STC molecules dissociate from the puncta following electron uptake, it would support the proposed model. On the contrary, the authors suggest that the reduction reaction possibly triggers FccA/STC punctum formation rather than dissociation, as supported by the additional experiment demonstrating that electron donation from CymA to STC is necessary for colocalization of STC with CymA mentioning that “Remarkably, STCPmC maintains a homogenous spatial distribution (Fig. 3c), while CymAGFP still formed puncta (Fig. 3d)”. This, I think, further undermines the claim that these proteins can readily dissociate and diffuse to the outer membrane.

Additionally, the authors have added the following in the main text that “Thus, while FccA and STC may not independently form condensates or mix into CymA condensates, their spatial proximity to CymA puncta is mediated through direct protein-protein interactions. This interaction-driven colocalization effectively brings the electron transfer partners into proximity.” While this explanation is plausible and consistent with known protein-protein interactions, the specific novelty of this statement is somewhat difficult to discern. If the main point is that direct interactions between FccA/STC and CymA contribute to efficient EET, this mechanism has already been well established in previous studies. As the authors themselves acknowledge, complex formation between these components is a canonical feature of the *Shewanella* EET pathway. Therefore, if the punctum formation does not involve additional intermolecular interactions among FccA or STC themselves, it remains somewhat unclear how puncta formation contributes to enhanced geometric proximity or improved electron transfer efficiency. Clarifying whether and how the spatial organization introduced by puncta formation provides functional advantages would help better establish the physiological relevance of this phenomenon.

Comment on Reply 1–4:

The authors added the following sentence that “Most CymAPmC puncta were observed in cells within 4 h during which there was no substantial new synthesis of CymAPmC in the cell (Supplementary Notes 3).” However, I was unable to confirm this statement from Supplementary Notes 3. While it is clear that CymAPmC concentrations did not increase substantially within 4 hours, the data do not provide direct evidence that most puncta were observed within this 4-hour window. If the authors wish to make this claim, they should provide data on the number of puncta per cell over time in Supplementary Notes 3 to substantiate it.

(Remarks on code availability)

As mentioned, I did not review the code, because there were no comments and replies related to the code for this revision.

Reviewer #2

(Remarks to the Author)

I appreciate the additional clarifications and new data provided. Some points are now clearer, but I still have some questions that could further improve clarity and depth of the work.

Reversibility.

- It remains unclear why the reversible nature of the puncta could not be examined using the single-cell assay. From Supplementary Figure 25, if the potential is interrupted, would the puncta dissipate or behave differently? By reducing imaging frequency, could the observation be extended, especially under the faster condition tracked in Figure 2?
- In Supplementary Figure 35, membrane localization is very clear in the control (panels a–c) but much less so in the “return” condition (panels g–i). Is this variability due to the specific example shown, or is it a general feature of CymA redistribution? Please clarify this variability in protein spatial distribution. Also in many images with puncta, membrane localization appears less clear. Adding a line profile orthogonal to the bacterium could help highlight and quantify this difference.

Spatial Description.

- On page 3, line 42, the term “spatial heterogeneity” remains ambiguous—does this refer to the percentage of cells with puncta, their shape, location, or size?
- The updated size quantification appears coarse; the threshold-based approach may obscure nanoscale features and limits assessment beyond the diffraction limit. While I understand this may be outside the main scope of the study, Supplementary Figure 6 raises further questions. In this figure, membrane localization of the clusters is difficult to discern. Given the clear membrane localization observed in the control widefield images, wouldn't the spatial resolution employed be expected to reveal a membrane outline? Please also clarify the effective spatial resolution of your imaging system, particularly for single-molecule tracking.

Dynamic profile.

- The FRAP data support the idea of biomolecular condensates, but currently appear as single, isolated examples. A broader, statistically representative dataset would strengthen this evidence.
- In Figure 3, the new data improve readability, but given the heterogeneous signal (some cells are very bright in the “red” channel, while other extremely dim), adding line profiles across cells could help confirm puncta presence and protein accumulation.

Minor points.

- The disposition of panel in figure 5 is confusing, considering the sudden change respect to the previous figures.

(Remarks on code availability)

Version 2:

Reviewer comments:

Reviewer #2

(Remarks to the Author)

The authors have satisfactorily addressed all of my previous comments. I recommend the manuscript for publication in its current form.

(Remarks on code availability)

[To all reviewers] Thank you very much for reviewing our manuscript and for your comments. We have now revised the manuscript accordingly; below are our point-by-point replies. All revisions in the main text and SI are marked in red fonts and cross-cited here. The text reproduced from the manuscript and SI in the replies below is indented and in black fonts with the red fonts marking the revisions. Again, we very much thank you for helping us improve.

Reviewer #1 (Remarks to the Author)

In this paper, the authors successfully combined electrochemical measurements with single-cell or single-molecule fluorescence microscopy. This innovative approach allows for the visualization of the aggregate formation of CymA, suggesting its link with electron flow. With the observation of the dynamics of CymA in individual cells and meticulous analysis, the study also demonstrates the heterogeneity of aggregate formation among different cells. The authors developed a model that colocalizes CymA aggregation with FccA/STC aggregation at the cell poles, facilitating efficient electron flow from the inside of the cell to the exterior. This model is fascinating and is particularly appealing in the context of the respiratory electron transport chain as well as the bacterial extracellular electron transfer (EET) pathway. While the biomolecules involved in the electron transport chain have been identified for some time, the dynamics and spatial organization of respiratory proteins have not been thoroughly explored due to technological limitations. This study highlights the significance of protein aggregate formation (possibly liquid-liquid phase separated domains) in the respiratory process. However, the relationship between aggregate formation and the proximity of CymA to FccA/STC is not sufficiently supported by the data presented, and this issue should be addressed prior to publication. Corrections and suggestions are provided below:

[Reply 1-0] Thank you very much for reviewing our manuscript and appreciating our work. Regarding your concern on the relationship between puncta formation and the proximity of CymA to FccA/STC, we have revised according to your comments below.

Major concerns:

1. In this paper, the formation of CymA, FccA, and STC aggregates at the cell pole was demonstrated, and a correlation between aggregate formation and extracellular electron transfer (EET) kinetics was suggested. The authors state that “the spatial colocalization of CymA’s direct electron-transfer partners, STC and FccA, which leads to geometric proximity and are thus expected to facilitate EET” (page 9). However, the connection between aggregate formation and the proximity of the proteins is not clear. This is because the aggregates consist of the same protein and do not involve mixing (no condensates containing both CymA and FccA/STC were observed). To clarify this relationship, I recommend quantitatively demonstrating the impact of aggregate formation on the distance between CymA and the nearest FccA/STC. This can be achieved by calculating the distances based on single-molecule tracking under four conditions: CymA (+, - condensate) versus FccA/STC (+, - condensate). Analyzing single-molecule tracking data would enable this quantification effectively.

[Reply 1-1] Thank you for your comments regarding the relationship between aggregate formation and the spatial proximity of CymA and FccA/STC. While it is correct that we did not directly observe mixed condensates containing both CymA and its periplasmic partners (FccA/STC), our model proposes a mechanism by which the experimentally-observed spatial colocalization can be achieved through direct protein–protein interactions. Specifically, in our model:

- 1) CymA becomes part of biomolecular condensates within inner membrane domains enriched in menaquinones (MQ), in which both CymA and MQ move dynamically. This dynamic motion within the spatially confined condensate facilitates electron transfer (ET) both among CymA molecules and between CymA and MQ.
- 2) Periplasmic cytochromes, such as FccA and STC, are known to interact directly with CymA through specific protein–protein interactions that mediate electron exchange during extracellular electron transfer (Ref. 15: Fonseca *et al.* Mind the gap: cytochrome interactions reveal electron pathways across the periplasm of *Shewanella oneidensis* MR-1. *Biochem. J.* **449**, 101–108 (2013); Ref. 16: Ross *et al.* Towards electrosynthesis in *Shewanella*: energetics of reversing the Mtr pathway for reductive metabolism. *PLoS ONE* **6**, e16649 (2011)). As these cytochromes approach the inner membrane, where CymA is anchored, they engage in these interactions to receive or donate electrons. As CymA undergoes spatial reorganization and forms puncta/condensates, FccA and STC that are complexed with CymA are recruited to these sites, resulting in their spatial colocalization with CymA condensates, as we observed experimentally (Fig 3).

Thus, while FccA and STC may not independently form condensates or mix into CymA condensates, their spatial proximity to CymA puncta is mediated through direct protein–protein interactions. This interaction-driven colocalization effectively brings the electron transfer partners into proximity, which we propose facilitates EET, as supported by the correlation between condensate formation and photocurrent shown in Fig. 4d. We have revised the writing in the text to clarify the above points (page 8 line 40-43 and caption in Supplementary Fig. 34):

..., CymA^{PmC} still formed puncta under anaerobic EET (Fig. 3g), ruling out FccA driving CymA punctum formation. Therefore, under anaerobic respiration, CymA forms puncta, which subsequently drives spatial colocalization of its direct electron-transfer partners STC and/or FccA in the periplasm. **Although FccA and STC may not independently form condensates or mix into CymA condensates, their spatial proximity to CymA puncta is mediated through direct protein-protein interactions^{15, 16}. This interaction-driven colocalization effectively brings the electron transfer partners into proximity, facilitating electron exchange for efficient EET. ...**

Supplementary Fig 34 | ... (iv) CymA's direct electron-transfer partners, STC and FccA, in the periplasm approach CymA in the condensate and interact dynamically with CymA **through direct protein-protein interaction^{15, 16}** to exchange electrons as part of EET pathways, showing colocalization. ...

We very much appreciate your suggestion to perform a two-color single-molecule tracking (SMT) experiment to simultaneously track CymA with either STC or FccA. However, this approach presents significant technical challenges:

Two-color SMT here requires the use of spectrally distinct photoactivatable fluorescent proteins (PA-FPs) that are also structurally stable in the periplasm. Due to the oxidizing conditions of the bacterial periplasm, fluorescent proteins containing cysteine residues are prone to disulfide bond formation, which compromises their folding and fluorophore maturation. Currently, there are no known pairs of distinct photoactivatable fluorescent proteins that are both color-distinguishable (e.g., red and green) and compatible with the periplasmic environment (i.e., lacking cysteines and remaining functional).

Even if such a pair of PA-FPs were available, it is challenging to measure the distance between interacting proteins in *live* cells. First, SMT relies on stochastic photoactivation and tracks one molecule at a time. It is not feasible to selectively activate and track two individual proteins that are part of the same complex or are spatially adjacent sequentially. For example, after imaging a single CymA molecule (randomly photoactivated from ~4000 copies), the probability of subsequently imaging/tracking its nearest STC or FccA partner is extremely low due to the stochastic nature of photoactivation. Second, because proteins are mobile in live cells, any subsequent tracking event may capture a different molecule or a previously interacting partner that has already moved away, making it impossible to reliably infer proximity or complex formation from 2-color SMT data alone.

Therefore, while conceptually appealing, two-color SMT is currently not feasible for measuring the spatial relationships between CymA and its periplasmic interacting partners *in live cells* under the constraints of the bacterial periplasm. To our knowledge, the best way to show protein colocalization in live cells remains two-color fluorescence imaging at diffraction-limited resolution, which we have implemented and presented in the manuscript (Fig. 3). Alternatively, one could employ two-color super-resolution imaging in *fixed* cells, where protein mobility and fluorescent protein maturation are no longer limiting factors, from which spatial relationship between two different proteins can be deduced at super-resolution. However, we approach the fixation-based methods with caution, as chemical fixation may introduce artifacts or alternative protein spatial distributions.

2. The EET process requires the shuttle of FccA/STC from the inner membrane to the outer membrane as mentioned on page 15. Aggregate formation of FccA/STC could hinder efficient diffusion in the periplasmic space due to the increased protein size and the reduced number of FccA/STC available for collision. Please discuss the potential disadvantages of aggregate formation on the efficient EET in the discussion part.

[Reply 1-2] Sorry for the confusion. It is not that FccA/STC forms aggregates, which then shuttle electrons between inner and outer membranes for EET. In our proposed model, the FccA/STC molecules that are spatially colocalized with CymA puncta are those forming complexes with CymA at the time when image was taken (Fig. 3). Within such FccA/STC-CymA complexes, electrons are exchanged readily. Such FccA/STC-CymA interaction complexes are expected to be dynamic, exhibiting transient associations and dissociations (Ref. 15: Fonseca et al. Mind the gap: cytochrome interactions reveal electron pathways across the periplasm of *Shewanella oneidensis* MR-1. *Biochem. J.* **449**, 101–108 (2013)). Therefore, this dynamic association/dissociation of STC/FccA with CymA, coupled with the rapid diffusion of free periplasmic components, should allow efficient ET between CymA on the inner membrane and the Mtr complex on the outer membrane, mediated by FccA and STC. Consistently, many FccA and STC proteins are observed not being colocalized with CymA, but free in periplasm, as supported by FccA/STC fluorescence outside the punctate regions (Fig. 3).

To clarify, we revised the mechanistic scheme in Supplementary Fig. 34 to add in some arrows to indicate diffusion of STC/FccA toward Mtr complex for ET and revised the caption as follows:

Supplementary Fig. 34 | ... (iv) CymA's direct electron-transfer partners, STC and FccA, in the periplasm approach CymA in the condensate and interact dynamically with CymA to exchange electrons as part of EET pathways, showing colocalization. **(v)** STC and FccA can subsequently **dissociate from the condensate and** diffuse toward the outer-membrane to exchange electrons with MtrABC/OmcA complex. ...

3. In this study, the author employed photocurrent measurement for assessing single-cell extracellular electron transfer (EET). While the idea and experimental setup are commendable, validating this system poses challenges. Although the authors provided detailed explanations of single-cell photocurrent measurement in supplementary notes 7 and 8, it remains difficult to conclude that the formation of CymA aggregates leads to enhanced EET kinetics. The photocurrent measurement using Cu₂WS₄ generates reactive oxygen species (ROS), and the presence of reactants may facilitate photocurrent generation. Additionally, biomolecular condensates can alter their redox properties, including their reactivity with ROS (Y. Dai et al, *Chem*, 2023, 9 (6), 1594). Thus, it is plausible that the formation of CymA aggregates simply enhances ROS consumption and photocurrent generation, rather than improving EET kinetics or the catalytic conversion of fumarate to succinate. To establish a clearer connection between CymA aggregation and EET, I recommend measuring ROS levels using appropriate indicators during the photocurrent measurement.

[Reply 1-3] We appreciate your thoughtful comments and the reference to recent literature highlighting the potential role of ROS in photocurrent generation.

In our study, ROS generation should be minimal under our photocathodic current measurements (i.e., reductive currents where photogenerated electrons in Cu₂WS₄ are donated out while the photogenerated holes are extracted away by the back-contact ITO electrode):

- 1) The photocurrent measurements were conducted under anaerobic conditions, eliminating molecular oxygen as a potential electron acceptor and thereby suppressing ROS formation via reduction pathways.
- 2) The efficient extraction of photogenerated holes through the ITO back-contact electrode was imposed by applying a sufficiently negative electrochemical potential, which prevents water oxidation and associated oxidative pathways for ROS generation. The experimental configurations are illustrated in Supplementary Fig. 1c.

Moreover, even if ROS generation or reductants (e.g., fumarate) contribute to photocurrent generation partly, it would be the same for off-cell regions or for on-cell without puncta. Therefore, the enhanced photocathodic current observed specifically in cells exhibiting CymA puncta cannot be attributed to general ROS reactivity or reactant availability.

Importantly, we observed no significant changes in electrochemical or physiological properties of the cell, which would occur if there were significant oxidative stress during the measurements: (i) the photocurrent enhancement remained stable over multiple cycles (10 s off, 15 s on, total 75 s) of light off/on illumination, and (ii) post-measurement imaging revealed no morphological changes, which would happen with cell lysis or death.

In addition, we acknowledge your point that biomolecular condensates may alter redox reactivity as noted in the cited work (Dai et al. *Chem* 9, 1594 (2023)). In this paper, the authors reported that the interface of biomolecular condensates can drive spontaneous redox reactions, including ROS generation in living cells. Per your suggestion, we measured ROS levels using Peroxy Orange 1 as a fluorescent indicator for hydrogen peroxide (H₂O₂), following Dai et al. (cited as Ref #96 in SI). We did not observe significant ROS generation within the biomolecular condensates (Supplementary Fig. 12, b-vs.-c and e-vs.-f). Instead, higher H₂O₂ levels were observed in inclusion bodies (i.e., insoluble aggregates) visible in bright-field transmission images (yellow arrows in Supplementary Fig. 12a and c). The observation of higher H₂O₂ in inclusion bodies agree with the cited work that interfaces between substantial phase differences may promote H₂O₂ generation. The fact that higher H₂O₂ was not observed in CymA condensates might stem from differences in the physical properties (i.e., liquidity) of the biomolecular condensates, which are protein-dependent. Therefore, in our study, CymA biomolecular condensate itself does not appear to generate significant ROS that could contribute to the measured photocurrent.

Still, it is conceivable that ROS generated by biomolecular condensates could contribute to the observed photocurrent. However, we believe this contribution is minimal for the following reasons:

- 1) Spontaneously generated ROS by biomolecular condensates is typically retained within the condensate. Both in vitro and in vivo experiments demonstrated that H₂O₂ predominantly localized inside the condensate (see Figs. 3 and 4 in Dai et al. *Chem* 9, 1594 (2023)). In the case of CymA in *Shewanella*, which is localized to the inner membrane, the spatial separation between the inner and outer membranes (~23.5 nm; measured by electron microscopy; Dohnalkova et al. Imaging hydrated microbial extracellular polymers: comparative analysis by electron microscopy. *Appl. Environ. Microbiol.* 77, 1254-1262 (2011)) makes it unlikely that ROS generated within the inner membrane-associated condensates would efficiently diffuse across membranes to reach the Cu₂WS₄ film electrode to significantly impact the photocurrent.
- 2) A previous study showed that through direct electrochemical measurements, ROS encapsulated inside stress granules contributed to just a few tens of pA (Hu et al. Electrochemical measurements reveal reactive oxygen species in stress granules. *Angew. Chem. Int. Ed.* 60, 15302-15306 (2021)). Although it is hard to directly compare with our experiments due to different biological systems, their reported current magnitude is an order of magnitude smaller than our measurement, -0.40 ± 0.07 (s.e.m.) nA.

All the above collectively led us to conclude that ROS generation in our system is likely insignificant and does not play a significant role in enhancing electron-uptake capabilities by cells. We have added the above discussion in the new Supplementary Note 8, including Supplementary Fig. 12 reproduced below.

Supplementary Fig. 12 | Reactive oxygen species level measurement in live *S. oneidensis*. Peroxy orange 1 (PO1) assay for the investigation of spatial distribution of H_2O_2 . **a-c**, Bright-field transmission (a) and wide-field epi-fluorescence images of ensemble-photoactivated anaerobically grown (lactate/ferric citrate) CymA^{GFP} strain (b, green channel), where generated H_2O_2 was assayed with PO1 (c, red channel). **d-f**, Same as a-c, but another region in the same experiment batch. Yellow arrows denote the inclusion body in the cells, identifiable in the bright-field transmission image (a). White arrows denote the CymA puncta, visible in CymA^{GFP} fluorescence images (b, e). Scale bars = 1 nm.

4. On pages 5-6, the authors mention that “We imaged the photoactivated CymAPmC in each cell over 6 h, during which there was no substantial new synthesis of CymAPmC in the cell (Supplementary Notes 3). After 6 h, CymAPmC puncta were observed in many cells (Fig. 2b, right, and Supplementary Fig. 20a, b).” The supplementary figure 5b in supplementary notes 3 shows CymA concentration in the cells across different images captured at 25-30 min intervals. They showed 8 data in anaerobically grown cells, thus, I think that no synthesis of CymAPmC is guaranteed only for 4 h at maximum.

[Reply 1-4] You are right. Sorry for not being precise. To clarify, we have revised the manuscript on page 6, line 1 as follows:

... We imaged the photoactivated CymA^{PmC} in each cell over 6 h. **Most CymA^{PmC} puncta were observed in cells within 4 h** during which there was no substantial new synthesis of CymA^{PmC} in the cell (Supplementary Notes 3). ...

5. Although cell division takes 6-8 hours under anaerobic conditions, as noted on page 13, some cells observed at the maximum of 4 hours in supplementary figure 5b should have undergone division (if doubling time is 8 hours, about half of the cells should divide in 4 hours). This is because not all cells synchronize their division timing. Given that cell division would dilute the matured CymAPmC concentration in each cell, it is puzzling why the concentration of matured CymAPmC remains unchanged after 4 hours of cultivation. A rational explanation for this observation is needed.

[Reply 1-5] As you pointed out, the cells are not synchronized in their division timing. In our experiments, cells were cultured anaerobically ahead of time at 30 °C for approximately two days to reach a steady-state phase, characterized by no significant further increase in optical density. Given that the typical doubling time under these culturing conditions is 6–8 hours (Ref. 51: Tang et al., Anaerobic central metabolic pathways in *Shewanella oneidensis* MR-1 reinterpreted in the light of isotopic metabolite labeling. *J. Bacteriol.* **189**, 894–901 (2007)), the population was expected to be in a relatively stable physiological state.

During our imaging (currently Supplementary Fig. 7b in this revised version), the temperature was at room temperature, which further slows cellular growth and division. Indeed, under these conditions, we did not observe any cell division or significant elongation over the typical imaging duration of ~6 h. Furthermore, this figure panel presents the average CymA concentration across a population of cells at each time point. Each image captures a heterogeneous population, with cells at various stages of the cell cycle—some already divided, others approaching division. Despite this variability, the average protein concentration among the population remained relatively constant over a 4 h period.

6. The authors demonstrate that the reduction of Fe(III) requires a lag time, which corresponds with the formation of CymA aggregates (see Figure 1). Along with their claim, it is suggested that the kinetics of extracellular electron transfer (EET) may be limited by the formation of these aggregates. However, accumulating evidence indicates that the acceleration of electron transfer in c-type cytochromes located on the outer membrane, such as MtrC and OmcA, is enhanced by flavins

and redox molecules, leading to an increase in EET kinetics by several-fold. This suggests that MtrC/OmcA may be the rate-limiting step in EET (E. Marsili et al., PNAS, 2008, 105(10), 3968; H. Canstein et al., Appl. Environ. Microbiol., 2008, 74(3)). Notably, this acceleration occurs rapidly (within several minutes) during the electron uptake process, which means it would not induce CymA aggregation and would primarily influence the electron flow across the outer membrane (A. Okamoto et al., Angew. Chem. Int. Ed., 2014, 53(41), 10988; Y. Tokunou et al., J. Phys. Chem. C, 2016, 120(29), 16168). This presents a conflict between the current data and traditional studies on this topic, warranting further explanation in the discussion part.

[Reply 1-6] Thank you for this thoughtful and important comment. We agree that previous studies have demonstrated that extracellular electron transfer (EET) kinetics can be significantly enhanced by flavins and other redox-active molecules, particularly through their interaction with outer membrane cytochromes such as MtrC and OmcA. These enhancements occur rapidly—within minutes—and primarily affect electron transfer across the outer membrane.

Our study focuses on a distinct step in the EET pathway: the electron transfer between menaquinones, CymA in the inner membrane, and the periplasmic proteins STC/FccA. We observed a lag phase in Fe(III) reduction that correlates with the formation of CymA condensates (Fig. 1), suggesting that this ET step may *initially* be rate-limiting. Once CymA forms condensates, this step becomes more efficient and likely no longer rate-limiting in the overall EET process.

Our findings and model do not contradict the findings of rapid EET enhancement via flavins at the outer membrane. Rather, it suggests a sequential model in which different steps in the EET pathway may be rate-limiting under different physiological or experimental conditions. In our measurements of temporal emergence of CymA puncta and Fe(III) reduction activity, cells were not pre-adapted to anaerobic conditions prior to measurement, and thus CymA condensates had not yet formed at the start of the experiment. This contrasts with the studies cited, where cells were typically pre-grown anaerobically, likely allowing CymA condensates to form prior to measurement. Under such conditions, the MtrC/OmcA-mediated ET step likely becomes rate-limiting, and flavin-mediated acceleration is readily observed.

Therefore, our findings complement existing literature by highlighting a previously underappreciated regulatory step in EET kinetics—CymA condensate formation—and its role in modulating the overall electron flow.

We have added a new paragraph discussing this point in the revised manuscript to better contextualize our results within the broader framework of EET regulation as follows (page 10, line 34-42):

Previous studies showed that in cells pre-adapted to anaerobic growth, flavins can enhance EET by accelerating electron transfer through outer membrane cytochromes like MtrC/OmcA, suggesting these as rate-limiting steps⁴⁰⁻⁴². In contrast, our data reveal that in transitioning from aerobic to anaerobic respiration, CymA puncta formation appears to be an upstream rate-limiting step that enables subsequent EET. Once CymA puncta are formed, the rate-limiting step can shift toward other downstream processes (e.g., those by Mtr/OmcA). Thus, our results complement existing literature and reveal a sequential regulation of rate-limiting steps in the overall EET pathway, which depends on the physiological state of the cell and the timing of protein spatial reorganization.

7. Although the authors suggested that menaquinone reduction drives CymA aggregate formation, the mechanisms underlying the FccA/STC aggregate formation are totally unclear. Please discuss possible models for this phenomenon.

[Reply 1-7] As noted in **Reply 1-2** above, FccA and STC do not form aggregates to shuttle electrons between the inner and outer membranes. Instead, these periplasmic cytochromes interact directly with CymA through specific and dynamic protein–protein interactions to facilitate electron exchange during extracellular electron transfer (Ref. 15: Fonseca et al. Mind the gap: cytochrome interactions reveal electron pathways across the periplasm of *Shewanella oneidensis* MR-1. *Biochem. J.* **449**, 101–108 (2013); Ref. 16: Ross et al. Towards electrosynthesis in *Shewanella*: energetics of reversing the Mtr pathway for reductive metabolism. *PLoS ONE* **6**, e16649 (2011)). Upon CymA condensate formation, FccA and STC dynamically localize near CymA puncta to enable efficient electron exchange, leading to their spatiotemporal colocalization.

Minor concerns:

1. The combination of iron metal and fumarate, used as an electron donor and acceptor (mentioned on page 3), is quite rare, I guess. Please include a reference that discusses this combination.

[Reply 1-8] Thank you for pointing this out. We added reference 25 in the manuscript on page 3, line 28 and the reference section, as follows:

... We also examined cells anaerobically grown with metal iron/fumarate as the electron donor/acceptor pair, where cells engage in EET through the same pathway but in the uptake direction²⁵...

25. Philips, J. *et al.* A novel *Shewanella* isolate enhances corrosion by using metallic iron as the electron donor with fumarate as the electron acceptor. *Appl. Environ. Microbiol.* **84**, e01154-18 (2018).

2. While photocurrent data is presented in Figure 4, the chronoamperometric data related to Figure 2 are missing. I recommend including this data in the Supplementary Information.

[Reply 1-9] Thank you for the concerns. We added the chronoamperometric data in Supplementary Fig. 27 and cross-cited it in Fig. 2 caption.

Supplementary Fig. 27 | Chronoamperometric data at a constant poised cathodic potential. Current vs. time at a constant -0.48 V vs. Ag/AgCl corresponding to Fig. 2. Due to the potentiostat's hardware/software limitations, current was measured for 900 s per run, and the poising potential was reapplied repeatedly for a total duration of 6 h. Inset: a magnified view of the first 10 s, highlighting the initial current drift and subsequent stabilization.

3. In this paper, Ag/AgCl is used as the standard potential, but the potential of Ag/AgCl changes according to the Cl⁻ concentration. Please state the Cl⁻ concentration somewhere (e.g. Ag/AgCl KCl sat., Ag/AgCl 3M KCl). (Remarks on code availability)

[Reply 1-10] Thank you for the comment. We used commercially available Ag/AgCl reference electrode (BASi, MW-2030), as described in the Supplementary Methods 1.6. This electrode contains 3 M NaCl. To clarify, we added 3 M NaCl in Supplementary Methods 1.6.

Reviewer #2 (Remarks to the Author):

The manuscript reports on the spatio-temporal coordination of protein complexes involved in extracellular electron transfer (EET) in the electroactive bacterium *Shewanella oneidensis*. Perk et al. integrate several experimental approaches to investigate the functional significance of the spatial organization of the hub protein CymA, particularly in relation to its direct electron-transfer partners. Ultimately, the authors attribute this reorganization to the formation of a biomolecular condensate.

Their use of a previously developed imaging platform to quantify the functional impact of spatial compartmentalization in electroactive bacteria is promising, as it seeks to directly link protein localization to functional outcomes.

[Reply 2-0] Thank you very much for your appreciation of this work and its potential future impact.

However, the manuscript emphasizes protein aggregation via biomolecular condensate formation as the central finding, yet the experimental data to support this conclusion are fragmented and partial. Several aspects of the study are reported at a superficial level, with key statistical details missing.

- Overexpression. What role does overexpression play in the observed spatial compartmentalization? The authors note that "overexpressing CymA shows higher EET activities" (Supplementary Fig. 25), and Figure 2g shows a correlation between CymA copy number and lag time. These observations suggest that expression levels may significantly influence both spatial organization and EET function. A more thorough discussion of this potential relationship would strengthen the interpretation of the results.

[Reply 2-1] We appreciate the reviewer's insightful comments regarding the potential role of overexpression in spatial compartmentalization and EET function.

First, we would like to clarify that our study does *not* involve overexpression of CymA and that CymA^{PmC} is expressed at physiological levels from its chromosomal locus. The statement in the previous Supplementary Fig. 25 (current Supplementary Fig. 12 in this revision) refers to others' work (Ref. 104: Vellingiri et al. Overexpression of c-type cytochrome, CymA in *Shewanella oneidensis* MR-1 for enhanced bioelectricity generation and cell growth in a microbial fuel cell. *J. Chem. Technol. Biotechnol.* **94**, 2115–2122 (2019)) where overexpression was shown to enhance EET activity. To clarify further in relation to your comments: We showed (Fig. 4d) that there is no significant correlation between total cellular CymA levels and enhanced photocurrent and there is only a marginal correlation ($p = 0.042$) with photocurrent density (current Supplementary Fig 13). On the other hand, enhanced photocurrent between cells that have puncta and without puncta showed significant differences even though these two types of cells have comparable cellular CymA concentration. Therefore, the presence of CymA alone *at its physiological expression level* is insufficient for EET; CymA puncta formation is essential to enable EET, suggesting that functional compartmentalization, rather than expression level within physiological ranges, governs EET efficiency.

While non-physiological overexpression could potentially influence both spatial organization and EET function, such conditions fall outside the scope of our current study. We agree that this is an interesting direction and have included a brief discussion in Supplementary Note 10, taking some previous text in the previous Supplementary Fig. 25 caption, along with the figure (now Supplementary Fig. 13 in the revision), to acknowledge this possibility and distinguish it from our study under physiological expression.

10. Discussion about CymA expression levels and EET function

While previous studies have reported that overexpression of CymA can enhance extracellular electron transfer (EET) activity¹⁰⁴, our study does not involve CymA overexpression. It is worth noting that all our observations were made under physiological expression levels. While previous studies have reported that overexpression of CymA can enhance extracellular electron transfer (EET) activity¹⁰⁴, our study does not involve CymA overexpression. It is worth noting that all our observations were made under physiological expression levels. We demonstrated that cellular CymA concentrations, either with or without puncta in cells, showed no significant correlation with their photocurrent Δi in Fig. 4d in the main text. In addition, we also investigated the correlation between CymA concentration and photocurrent density (Supplementary Fig. 13). Here the photocurrent density was calculated by dividing the single-cell current by the area of the cell in contact with the Cu₂WS₄ thin film within the laser-illuminated region (Fig. 4a). Consistent with the result in Fig. 4, cells with puncta showed significantly more negative current density compared to cells without puncta (Supplementary Fig. 13, right panel), even though their cellular concentration of CymA was similar (top panel), indicating again that CymA expression level alone is insufficient to drive EET. Instead, our data indicate that the formation of CymA puncta is a critical determinant of EET functionality. Interestingly, cells with puncta showed a marginal negative correlation, with Pearson's cross correlation coefficient of -0.28 with a P -value of 0.042 (Supplementary Fig. 13, solid red symbols), suggesting that more CymA in the puncta could potentially facilitate slightly better EET *at the per-contact-surface-area level*, which would be consistent with previous finding that cells

overexpressing CymA show higher EET activities¹⁰⁴. However, it's important to note that our study does not involve CymA overexpression—all observations here were made under physiological expression levels.

While it is plausible that non-physiological overexpression could influence both spatial organization and EET performance, potentially by altering protein localization dynamics or stoichiometry, such scenarios are beyond the scope of this study. Future work could explore how overexpression level perturbations affect spatial compartmentalization and EET efficiency.

Supplementary Fig. 13 | Marginal negative correlation between cellular concentration of CymA and photocurrent density of single cells.

Photocurrent density calculated by dividing the single-cell current by the area of the cell occupied on the Cu₂WS₄ thin-film within the laser illumination spot (Fig. 4a). Circles: individual cells containing puncta (solid red) or no puncta (open black). Denoted are Pearson's correlation coefficient ρ for cells with puncta (red) and cells without (black) and their P values (two-sided t-test: $P < 0.05$, significant). Top/right: Projected histograms. Statistical significance was determined using Welch's two-sample t-test (NS, not significant; $P < 0.05$, significant).

- Reversibility. This critical aspect is only briefly mentioned and not quantitatively addressed, despite its importance for identifying biomolecular condensates, as noted in the literature and by the authors themselves (page 13, lines 7–33). Specifically, the lack of quantification in Supplementary Figure 29 limits a full understanding of the reversibility process and its role in the observed phenomena. Moreover, the distribution shown in panels g–i appears more homogeneous compared to panels a–c in the same figure, raising further questions: how efficient is the reversibility? Can it be visualized in the same sample over time to track the process dynamically? Additionally, what is the viability of the cells after the assay? On page 3, line 25, how does reversibility impact the quantification of the number of puncta? In the electrochemical manipulation assay, the system remains under anaerobic conditions, so what is the nature of the 10% reversibility mentioned? **[Reply 2-2]** Thank you for your invaluable and thoughtful comments on reversibility of puncta formation.

To better characterize the reversibility of puncta formation, we quantified the fraction of cells displaying puncta under three different conditions: aerobic, anaerobic, and re-aerated (following anaerobic exposure). Under anaerobic conditions, $42 \pm 14\%$ of cells exhibited puncta initially. Upon returning to aerobic conditions, this number dropped to $5.4 \pm 3.8\%$, which is similar to the fraction that were initially grown under aerobic condition ($4.9 \pm 6.1\%$), indicating that most of the puncta-positive cells lost puncta during re-aeration—an 88% reversal rate. These results are now presented in an additional panel in Supplementary Fig. 35j.

To further support this observation, we analyzed the skewness of fluorescence pixel intensity distributions of individual cells under the same three conditions. The average skewness value among the cells returned to near-aerobic levels after re-aeration (new panel Supplementary Fig. 35k), consistent with the visual similarity between panels g–i and a–c in Supplementary Fig. 35.

Supplementary Fig. 35 | When anaerobically grown cells with CymA^{PmC} puncta were re-introduced to aerobic conditions, CymA became uniformly distributed across the cell envelope. a-c, (a) Transmission image and (b) wild-field fluorescence image of CymA^{PmC} cells grown aerobically with amino acids and oxygen. (c) Zoom-in of dotted boxes in (b). **d-f,** (d) transmission image and (e) wild-field fluorescence image of CymA^{PmC} cells grown anaerobically with lactate and ferric citrate, exhibiting CymA puncta. (f) Zoom-in of dotted boxes of (e). **g-i,** (g) Transmission image and (h) wild-field fluorescence image of anaerobically grown CymA^{PmC} cells re-introduced to aerobic condition and grow for 8 h. CymA puncta disappeared, and CymA became uniformly distributed across the cell envelope. (i) Zoom-in of dotted boxes of (h). Scale bars: 3 μm . **j,** Fraction of cells showing CymA^{PmC} puncta grown condition corresponding to (a-c), (d-f), and (g-i) for each column. 1st and 2nd column data are same as Fig. 1f. For 3rd column, total 462 cells in different 10 images were counted. Open circles: fractions of cells with puncta from individual images. Means and error bars (s.d.) are from multiple images from 2-3 biological replicates, with the calculations weighed by the cell counts. **k,** Skewness corresponding to three conditions in (j) Means (open circle) and error bars (s.d.).

We agree that tracking the same set of cells over time while switching between aerobic and anaerobic conditions would be great. Yet, it is technically challenging, requiring switching from aerobic to anaerobic solutions and imaging over long time. Transitioning from aerobic to anaerobic environments requires strict oxygen control, which our current flow-cell set up unfortunately cannot provide. Also, anaerobic growth conditions typically either use solids (e.g., metal iron or ferric oxide) or generate precipitates (e.g., the ferrous citrate precipitate from ferric citrate reduction), hindering fluorescence imaging as mentioned in the manuscript in page 5, line 30. Additionally, long-term fluorescence imaging over 2-to-3 different growth condition phases essentially double/triple the observation time (each phase would need 8 hours), for which fluorescence photobleaching will be too substantial. Due to these limitations, we chose to culture cells under defined conditions for these experiments and image them at the endpoint, ensuring consistent and interpretable data.

After switching anaerobic condition to aerobic condition, OD number of cells increased, indicating that cells are viable with active division and growth. In addition, previous studies on *S. oneidensis* support the viability of cells grown

during or after anaerobic conditions (Roy et al. Applied electrode potential leads to *Shewanella oneidensis* MR-1 biofilms engaged in direct electron transfer. *J. Electrochem. Soc.* **160**, H866 (2013); Ref. 5: Cao et al. Silver nanoparticles boost charge-extraction efficiency in *Shewanella* microbial fuel cells. *Science* **373**, 1336-1340 (2021); Yu et al. Conductive artificial biofilm dramatically enhances bioelectricity production in *Shewanella*-inoculated microbial fuel cells. *Chem. Commun.* **47**, 12825-12827 (2011)).

Quantification of fraction of cells showing puncta was performed using *static* image snapshots of many cells at a single time point after growth. These measurements represent steady-state distributions and do not reflect dynamic changes over time.

In the time-course electrochemical measurements under anaerobic conditions, we observed that in approximately 10% of cells, the formed puncta dissipated over time during imaging, even though the environmental conditions remained unchanged. Based on our hypothesis that puncta likely form within membrane microdomains enriched in menaquinone (MQ), fluctuations of MQ levels or membrane composition could *occasionally* destabilize these microdomains, leading to puncta dissolution. However, this appears to be a relatively rare event, in contrast to active environmental manipulation switching from anaerobic to aerobic conditions, which resulted in a much higher degree of reversibility, with 88% of cells with puncta lost their puncta (Supplementary Fig. 35j).

- Dynamic exchange. In Supplementary Note 11, no clear difference in the topological distribution of the two diffusive populations is visible in the images presented in Supplementary Figure 12. The population with higher diffusivity appears less dense, likely due to its smaller fraction, but there is no apparent spatial separation that would support the hypothesis that “raft-like lipid domains, formed with menaquinone and the membrane, are involved in the formation of CymA puncta.” This observation alone does not sufficiently support the proposed model.

[Reply 2- 5] In Supplementary Fig. 17 (previous Supplementary 12), we showed our observation that under aerobic conditions, CymA^{PmC} were homogeneously distributed over the cell envelope, for both CymA dominantly in the D_1 state (slow diffusion) and those dominantly in D_2 state (faster diffusion) (panel b and c). In contrast, under anaerobic conditions (e.g., with lactate/ferric oxide), CymA^{PmC} dominantly in the D_1 state were primarily located at the punctum (panel e), while CymA^{PmC} dominantly in the D_2 state exhibited largely homogeneous distribution across the cell envelope (panel f). Based on the extracted diffusion constants, CymA molecules in the D_1 state correspond to the inner membrane-anchored fraction and those in the D_2 state correspond to the periplasmic fraction.

To quantify the topological distribution, we have now examined the distribution of pairwise distances between the first locations of each tracked CymA dominantly in the D_1 state and the D_2 state, respectively. For CymA^{PmC} grown under aerobic conditions, the mean values of pairwise distance distribution from locations dominantly in the D_1 state (inner-membrane-anchored fraction) and D_2 state (periplasmic fraction) were 400 and 470 nm, respectively (new Supplementary Fig. 17d). On the other hand, for CymA^{PmC} grown under anaerobic conditions, the mean values of pairwise distance distribution of dominantly in the D_1 and D_2 states are 270 and 450 nm, respectively, showing a much larger difference (new Supplementary Fig. 17h).

Therefore, the CymA molecules in the slower D_1 diffusion state (CymA in the membrane) change their spatial distribution from aerobic growth to the anaerobic growth during EET, while those in the faster D_2 diffusion state (i.e., CymA in the periplasm) do not change their spatial distribution. These results indicate that CymA’s spatial distribution changes are likely linked to membrane reorganization, which we proposed/hypothesized to be raft-like domain formation, due to menaquinone level increase in the membrane, which led to the formation of CymA^{PmC} puncta.

We agree that the observation in Supplementary Fig 17 (previous Supplementary Fig. 12) *alone* does not sufficiently support our proposed model. Such observation is just one piece of evidence supporting our model, which comes from a collection of evidence: (1) temporal correlation of MQ upregulation and puncta formation (Fig. 5h), (2) MQ-rich domains known to form raft-like domains (Ref. 49: Kellermann et al. Important roles for membrane lipids in haloarchaeal bioenergetics. *Biochim. Biophys. Acta* **1858**, 2940–2956 (2016); Ref. 56: Snead et al. Membrane surfaces regulate assembly of ribonucleoprotein condensates. *Nat. Cell Biol.* **24**, 461–470 (2022); Ref. 57: Wang *et al.* Coupling of protein condensates to ordered lipid domains determines functional membrane organization. *Sci. Adv.* **9**, eadf6205 (2023)). (3) CymA remains mobile in the puncta (Fig. 5), consistent with the fluidic nature of raft-like lipid domains.

As discussed above, we have added pairwise distance distribution analysis to Supplementary Note 13 and Supplementary Fig. 17 and revised it as shown below:

.... To quantify the topological distribution, we examined the distribution of pairwise distances between the first locations of each tracked CymA dominantly in D_1 state and D_2 state, respectively. For CymA^{PmC} grown under aerobic conditions, the mean values of pairwise distance distribution from locations dominantly in the D_1 state (inner-membrane-anchored fraction) and D_2 state (periplasmic fraction) were 400 and 470 nm, respectively (Supplementary Fig. 17d). On the other hand, for CymA^{PmC} grown under anaerobic conditions, the mean values of pairwise distance

distribution of dominantly in the D_1 and D_2 states are 270 and 450 nm, respectively, showing a much larger difference (Supplementary Fig. 17h).

Therefore, the slower D_1 diffusion state (CymA in the membrane) changes their spatial distribution from aerobic growth to the anaerobic growth during EET, while the faster D_2 diffusion state (CymA in the periplasm) does not change their spatial distribution. These results indicate that CymA's spatial distribution changes are likely linked to membrane reorganization, which we proposed/hypothesized to be raft-like domain formation, due to menaquinone level increase in the membrane, which led to the formation of CymA^{PmC} puncta.

Supplementary Fig. 12 | Reconstructed super-resolution images of CymA in two diffusion states. **a,e**, PDF(r) of first displacement lengths from the CymA^{PmC} strain (a) grown under aerobic conditions, with a dashed yellow line marking the threshold (r_0) at 311 nm, and (e) grown under anaerobic conditions (Lactate/Ferric oxide), with the threshold at 255 nm. Displacement length r lower than the threshold encompass 99% of the dominant D_1 state (slow diffusion state) displacements in the original PDF(r). (a) and (e) are same figures as Supplementary Fig. 11a and d. **b, c, f, g**, Reconstructed super-resolution images of CymA^{PmC} molecules that are dominantly in the (b, f) D_1 diffusion state or dominantly in the (c, g) D_2 diffusion state, corresponding to (b, c) aerobic and (f, g) anaerobic growth conditions. (f) is the same figure as Fig. 5d in the main text. Scale bars = 1 μ m. **d**, Pairwise distance distribution of first locations of single CymA^{PmC} tracks shown in panel b (yellow line) and c (pink line). **h**, same as d but the tracked CymA^{PmC} molecules corresponding to panel f (yellow line) and g (pink line). Black vertical lines, mean value of the distribution.

Additionally, in Supplementary Note 11.2, the percentages reported lack associated error estimates, making it difficult to assess the robustness of the data. How large is the statistical sample used in this analysis?

[Reply 2-6] Sorry for not providing the error bars earlier. We added error bars of fractional population from displacement distribution fitting of Brownian model (Supplementary Note 11.2, now 13.2 in the revision). We also added the total number of cells we analyzed and the number of tracked CymA as follows in page S30, line 30-31:

...(a total of 20,273 CymA^{PmC} were tracked across 268 cells, with 2,918 localized within puncta and 17,355 outside puncta).

Could the periplasmic fraction be quantified more rigorously? Would super-resolution techniques such as SMLM provide a better 3D characterization of CymA distribution? While the authors attempt this in Figure 5a–c, it is unclear how 3D diffusion affects the thresholding applied to classify step sizes. How robust is the classification method used to sort diffusion steps? Simulations of various diffusion modes within a realistic bacterial geometry could help evaluate and validate the applied analysis pipeline.

[Reply 2-7] Thank you for the suggestions regarding the quantification and spatial characterization of CymA distribution.

To quantify the periplasmic fraction, biochemical methods such as cell lysis followed by enrichment of periplasmic proteins can be used. While they can estimate the periplasmic fraction, these approaches are relatively crude and require large-scale cultures, lacking the spatial resolution and quantitative precision. In addition, as you suggested, 3D single-molecule localization microscopy (SMLM) could, in principle, provide improved spatial resolution. However, even with

SMLM’s typical resolution of 5–10 nm, distinguishing between the inner membrane (IM) and periplasmic localizations remains challenging due to their close proximity. Our wide-field fluorescence images (Fig. 1b, Supplementary Fig. 21) already demonstrate clear cell-envelope localization of CymA, supporting its presence in the IM or periplasmic space.

To resolve the IM and periplasmic fractions, we relied on differences in molecular mobility. Specifically, by analyzing the single-molecule displacement length distributions, we identified two diffusion states, with diffusion constants D_1 and D_2 , corresponding to distinct populations with fractional amplitudes A_1 and A_2 . These two diffusion states are assignable to inner-membrane anchored proteins and periplasmic proteins, allowing us to infer spatial locations.

In Fig 5c and Supplementary Note 13, we used displacement length thresholding to separate the slower-diffusing population (D_1 state). This thresholding approach is only an approximation, as we stated in the manuscript, to select a population that is dominated by the slower-diffusing state.

We acknowledge that our single-molecule tracking data are based on 2D projected displacements, which inherently differ from the original 3D diffusion. To deconvolute this projection-induced effects, we applied an inverse transformation method to deduce the 3D displacement distribution from 2D data, provided the exact cell geometry that was measured experimentally from optical transmission images, as described in Supplementary Note 12. This approach was validated in previous studies using both simulations and experimental data of multi-state diffusion in bacterial cells for both cytosolic and cell envelope protein diffusions (e.g., Oswald et al. Imaging and Quantification of Trans-Membrane Protein Diffusion in Living Bacteria. *Phys. Chem. Chem. Phys.*, **16**, 12625–12634 (2014); Chen et al. Quantifying multistate cytoplasmic molecular diffusion in bacterial cells via inverse transform of confined displacement distribution. *J. Phys. Chem. B* **119**, 14451–14459 (2015); reference 107 and 43).

The discussion of diffusion coefficients is also confusing. In one instance (page 13, lines 25–26), the diffusion is described as “slightly slower,” whereas in another (page 12, line 12), it is “similar/slightly slower.” This inconsistency, along with the lack of statistical replicates (how many independent experiments were performed?), makes it difficult to assess whether a meaningful change in diffusivity is present. If, the diffusion coefficient is confirmed to remain unchanged, the cited literature only partially supports the authors’ conclusion that this is indicative of a biomolecular condensate. In the referenced study (page 13, lines 26–29), the lack of change in diffusion was observed only after correcting for centroid tracking of the aggregate. Has a similar correction been applied here?

[Reply 2-8] We are sorry for the inconsistency.

We used the phrase “similar/slightly slower” to encompass the range of diffusion behaviors observed under various anaerobic growth conditions, including both electron uptake and electron outflow scenarios, in comparison with that under aerobic growth. Please see the details in Supplementary Table 4 below. To improve clarity, we have revised the manuscript to describe the diffusion as “slightly slower” consistently with a quantified reduction of approximately 20–33%, except under the anaerobic condition with hydrogen/fumarate, where the D_1 value falls within the error margins and is statistically indistinguishable from that under aerobic growth. For statistical replicates, we confirm that all diffusion measurements were performed with 2-3 biological replicates, as noted in Fig. 1 caption.

Supplementary Table 4 | Diffusion constants and fractional populations of CymA grown under various growth conditions.

	Growth condition	D_1 ($\mu\text{m}^2/\text{s}$)	A_1	D_2 ($\mu\text{m}^2/\text{s}$)	A_2	copy number	[CymA] (μM)	No. of cells
1	AA/O ₂	0.15 ± 0.01	0.74 ± 0.02	5.35 ± 0.55	0.26 ± 0.02	2952 ± 1002	4.8 ± 1.8	269
2	Lactate/Ferric citrate	0.12 ± 0.01	0.76 ± 0.02	7.36 ± 0.55	0.24 ± 0.02	3770 ± 1506	5.6 ± 2.6	481
3	Fe/Fumarate	0.10 ± 0.01	0.80 ± 0.02	5.37 ± 0.83	0.20 ± 0.02	2900 ± 1680	5.2 ± 2.8	272
4	Lactate/Ferric oxide	0.10 ± 0.01	0.82 ± 0.01	4.29 ± 0.59	0.18 ± 0.01	5512 ± 2360	6.6 ± 2.6	121
5	H ₂ /Fumarate	0.15 ± 0.01	0.78 ± 0.02	7.67 ± 1.16	0.22 ± 0.02	4802 ± 1970	4.6 ± 1.4	167

We appreciate the reviewer’s thorough review regarding diffusion coefficient in the context of biomolecular condensates. Our most important point here is that even after condensate formation, proteins are still mobile inside the condensate, instead whether the diffusion coefficient is *unchanged or slightly reduced*. Both scenarios have been reported in the literature: some studies observe no significant change in diffusion upon condensate formation (e.g., for the protein HslU: Jin et al. Membraneless organelles formed by liquid-liquid phase separation increase bacterial fitness. *Sci. Adv.* **7**, eabh2929 (2021)), while others report a modest reduction, varying by proteins’ properties (e.g., for the proteins PopTag^{SL}, PopTag^{LL}, McdB: Hoang et al. An experimental framework to assess biomolecular condensates in bacteria. *Nat. Commun.* **15**, 3222 (2024).). To avoid confusion, we have added a reference (ref 50) and clarified this point in the revised text (page 13, line 31-33).

... The fact that the diffusion coefficient of the dominant mobile state of CymA becomes just slightly slower upon forming condensates is also in agreement with literature: the apparent diffusion coefficient of HslU (ATP-dependent protease) remained similar from $0.12 \pm 0.07 \mu\text{m}^2/\text{s}$ (early stage) to $0.10 \pm 0.03 \mu\text{m}^2/\text{s}$ (late stage) upon forming condensate⁵⁴; PopTag^{SL} and PopTag^{LL}, (polar organizing protein Z) and McdB (maintenance of carboxysome distribution protein B) displayed a modest reduction, varying by proteins' properties⁵⁰.

For correcting for the centroid of the aggregate, in our case, the puncta exhibited no discernible diffusion ($0.002 \pm 0.001 \mu\text{m}^2/\text{s}$, i.e., stationary) during the same period of SMT acquisition (1.8 s for each molecule). 209 puncta were tracked by open-source plugin in image J, TrackMate (Ershov et al. TrackMate 7: integrating state-of-the-art segmentation algorithms into tracking pipelines *Nature methods* **19**, 829-832, (2022)). Thus, following the procedure described in Ref. 54 (Jin et al. Membraneless organelles formed by liquid-liquid phase separation increase bacterial fitness. *Sci. Adv.* **7**, eabh2929 (2021)) by subtracting the diffusion coefficient of puncta from the SMT diffusion coefficient, the corrected diffusion coefficient remains effectively *unchanged* considering the error bars.

It is important to note that SMT was performed after overnight anaerobic growth, at which point puncta were already formed. The SMT acquisition lasted ~15 min, which is significantly shorter than the timescale of puncta formation or evolution (~4 h). Therefore, the SMT data reflect steady-state behavior, and are not confounded by the condensate formation dynamics during imaging. As discussed, we have revised (page SI 27, line 22-25).

... Note that CymA puncta exhibited no discernible diffusion (with an apparent diffusion coefficient of $0.002 \pm 0.001 \mu\text{m}^2/\text{s}$, i.e., effectively stationary) during the same period of CymA SMT acquisition (1.8 s for each molecule), thus the corrected single-molecule CymA diffusion coefficient remains effectively unchanged considering the error bars (Supplementary Fig. 16).

Supplementary Fig. 16 | CymA^{PmC} puncta are effectively stationary in *S. oneidensis*. a. Representative fluorescent image of CymA^{PmC} in *S. oneidensis* grown anaerobically with iron/fumarate, overlaid with tracked puncta (purple circle). b, Magnified view of selected cells with tracks in the puncta. c, Histogram of the apparent diffusion coefficient D of CymA^{PmC} puncta. A total of 209 puncta were tracked by open-source image J plugin, TrackMate¹⁰⁸ Mean \pm s.d. = $0.002 \pm 0.001 \mu\text{m}^2/\text{s}$.

Overall, to more convincingly establish the biomolecular condensate identity of the CymA assemblies, a photobleaching assay (e.g., FRAP) should be incorporated into the validation strategy, in line with established approaches in the field.

[Reply 2-9] Per your suggestion, we have performed fluorescence recovery after photobleaching (FRAP) measurement (new Supplementary Fig. 31 below). Using a laser beam of ~1 μm in lateral size, a single CymA punctum was photobleached for 10 s. Then, we measured the fluorescence intensity at 90 s intervals for up to 9 min afterward (panels in a). Fluorescence in the bleached region recovered with a half time of 3.4 min to approximately 76% of the initial intensity at steady state (panel b), indicating that CymA molecules can undergo dynamic exchange between the puncta and surrounding. Compared with single molecule tracking data, the apparent timescale here is much slower, but it might be due to the limited diffusion from one side the cell (the original punctum was near a cell pole), taking longer time to recovery. This delayed diffusion agrees with the previous study (Ref. 54: Jin et al. Membraneless organelles formed by liquid-liquid phase separation increase bacterial fitness. *Sci. Adv.* **7**, eabh2929 (2021)). We added FRAP measurement and the data analysis in (Supplementary Fig. 31) and mentioned in the manuscript in page 12 in line 30-31.

... Additional fluorescence recovery after photobleaching (FRAP) measurements further confirmed the dynamic exchange of CymA (Supplementary Fig. 31)....

Supplementary Fig. 31 | Fluorescence recovery after photobleaching (FRAP) measurement. **a**, Representative epifluorescence images of *S. oneidensis* cells pre-grown anaerobically with lactate/ferric citrate. Using a laser beam of $\sim 1 \mu\text{m}$ in lateral size (dotted line), a CymA punctum was photobleached for 10 s. Then, the fluorescence images were recorded at 90 s intervals for up to 9 min afterward. Scale bars: $1 \mu\text{m}$. **b**, Normalized fluorescence intensity is fitted with an exponential rise function. The fluorescence in the bleached region recovered with a half time of 3.4 min to approximately 76% of the initial intensity at steady state, indicating that CymA molecules can undergo dynamic exchange between the puncta and the surrounding. Compared with single molecule tracking data, the apparent timescale is much slower, but it might be due to the limited diffusion from one side the cell (the original punctum was near a cell pole), taking longer time to recover, consistent with the previous study⁵¹.

Other major comments

- Pag. 1-line 27-28, pag 3 line 5-6 and pag. 15 line 22-34. The authors close both the abstract and the conclusion paragraph with a digression on the relevance of biomolecular condensate that sounds vague and difficult to link to the directly presented work. What do the authors specifically mean when they say that the observation could have a “broad relevance to the function of biomolecular condensates in biology”. A clearer description of why the condensate is important for the specific system would give a better closure to their finding and more easily identify the context and scope of the work.

[Reply 2-10] Thank you for your suggestion. We meant that our finding adds to the growing body of evidence that biomolecular condensates are not merely passive organizational structures but can actively contribute to key biochemical processes. By demonstrating a direct functional role in EET (a new discovery here), our work suggests that the relevance of condensates may extend beyond previously recognized contexts, potentially influencing a broader range of biological functions. We have now revised the Abstract, Introduction, and Conclusion sections to more clearly articulate the specific functional role that the condensate plays in facilitating extracellular electron transfer (EET) in our system.

Abstract section: ...These orchestrated spatiotemporal protein dynamics **extend the functional roles of biomolecular condensates to include facilitation of EET in bacteria, with broader implications for cellular processes.**

End of introduction: ...Altogether, our findings provide new insights into how cells coordinate the spatiotemporal actions of proteins that reside in different cellular compartments to control EET **and may extend the functional roles of biomolecular condensates to include facilitation of EET in bacteria, suggesting a broader relevance of condensates in cellular function.**

Discussion: ...Quantifying the spatiotemporal dynamics of these proteins in cells in correlation with their cellular function of EET **at the single-cell level suggests that the relevance of condensates may extend beyond previously recognized contexts, potentially influencing a broader range of biological functions.**

- Pag. 3 line 23. The authors conclude that the aggregation is diffraction-limited, and at this level of spatial precision, all the conditions result in the same aggregation. Since they have the capability to perform single molecule localization microscopy in their integrated platform and with the fluorescent protein tag used, would super resolution provide a finer description of this association?

[Reply 2-11] Thank you for the comment. We would like to clarify that while the observed puncta appear as $\sim 400\text{--}600 \text{ nm}$ in size under diffraction-limited imaging, this does not imply that the physical size of puncta themselves are diffraction-

limited. Rather, it indicates that the size of the puncta exceeds the resolution limit of fluorescent microscopy, ~200–300 nm. While our single-molecule tracking (SMT) offers higher spatial resolution, the puncta are already larger than the diffraction-limited resolution, and thus super-resolution imaging would result in similar size measurements.

Per your suggestion, we further analyzed our single-molecule tracking data to determine the puncta size from the super-resolution images. First, we reconstructed super-resolution image as explained in supplementary Note 13. Then, it is transformed to 2D histogram, and the puncta are classified from the 2D histogram by the same machine learning algorithm. The puncta diameter obtained from such super-resolution image analysis is $0.59 \pm 0.45 \mu\text{m}$ on average (Column 2 in Supplementary Table 3), similar to the result from analyzing diffraction-limited images ($0.62 \pm 0.49 \mu\text{m}$ on average; Column 1 in Supplementary Table 3).

Moreover, we reduced the bin size to approximately the single-molecule localization uncertainty (~40 nm) to investigate whether finer structural features could be resolved within the puncta in the super-resolution images (Supplementary Fig. 6d-f). However, no additional substructures were observed, which is not surprising, as possible protein spatial distribution within each condensate punctum would be beyond the resolution of super-resolution imaging.

We have added above analysis and new figure in the Supplementary Note 2.

Moreover, puncta sizes are determined from super-resolution images as well. First, we reconstructed super-resolution images as explained in Supplementary Note 13. Then, it is transformed to 2D histograms (Supplementary Fig. 6b), and the puncta are classified from the 2D histogram image by the same machine learning algorithm (Supplementary Fig. 6c). Puncta diameter obtained from such super-resolution images is $0.59 \pm 0.45 \mu\text{m}$ on average (Column 2 in Supplementary Table 3), similar to the result from analyzing diffraction-limited images ($0.62 \pm 0.49 \mu\text{m}$ on average; Column 1 in Supplementary Table 3). To investigate whether finer structural features could be resolved within the puncta in the super-resolution images, we further reduced the super-resolution image bin size to approximately the single-molecule localization uncertainty (~40 nm. Supplementary Fig. 6d-f). However, no additional substructures were observed, which is not surprising, as possible protein spatial distribution within each condensate punctum would be beyond the resolution of super-resolution imaging. All results are summarized in Supplementary Table 3.

Supplementary Fig. 6 | Determination of CymA^{PmC} puncta size from reconstructed super-resolution image. **a**, Representative reconstructed super-resolution image from single molecule tracking data. Details are in Supplementary Note 12. **b**, 2D histogram image corresponding to (a). Bin size = 135 nm. **c**, Binary image from (b) illustrating puncta classified by the algorithm. **d**, **e**, **f**, Exemplary 2D-histogram image reconstructed from single molecule tracking data. Bin size = 34 nm. All scale bars = 1 μm .

Supplementary Table 3 | Size of CymA^{PmC} puncta in cells grown anaerobically with different electron donor/acceptor pairs. Diameters: mean \pm s.d.

	Electron donor/acceptor pair	Puncta diameter (in μm) from pixel area – diffraction limited image	Puncta diameter (in μm) from pixel area – super resolution image	Puncta diameter (in μm) from circle fitting
1	H ₂ /Fumarate	0.63 ± 0.52	0.57 ± 0.45	0.44 ± 0.14
2	Lactate/Ferric citrate	0.57 ± 0.43	0.57 ± 0.46	0.42 ± 0.12

3	Fe/Fumarate	0.58 ± 0.43	0.62 ± 0.44	0.44 ± 0.16
4	Lactate/Ferric oxide	0.67 ± 0.51	0.60 ± 0.48	0.50 ± 0.18
Average		0.62 ± 0.49	0.59 ± 0.45	0.44 ± 0.16

- Pag. 6 line 27. How does photobleaching affect the assay reported in Figure 2? Could it introduce a bias, particularly over longer time scales? If the emergence of a skewed distribution is being used as an indicator of aggregation, how robust is the temporal characterization—specifically the sigmoidal fit used by the authors? From the examples provided, it appears that the dynamics of aggregate formation may occur on a faster timescale than that captured by the current timelapse acquisition. This could potentially explain the lack of correlation observed for the transition time. This interpretation is further supported by Supplementary Figure 21b, where the transition times are heavily skewed toward zero. Would a faster imaging rate allow for a more accurate temporal description of the aggregation process? Improved time resolution might clarify the onset and progression of puncta formation and enhance the reliability of the fitted kinetic models.

[Reply 2-12] Thank you for this thoughtful and important comment.

We acknowledge that photobleaching imposes a limitation on our ability to achieve both high temporal resolution and long-term time-lapse imaging. To address this, we optimized our imaging conditions to balance these competing demands, settling on a 15 min interval that allowed us to capture the overall dynamics of puncta formation while minimizing photodamage. As you correctly points out, many of the extracted transition times are shorter than 0.5 h, which approaches the limit of our time resolution. As a result, some transition times may be overestimated (i.e., not fully resolved).

Nevertheless, our main conclusions remain valid: (1) the transition time is significantly shorter than the lag time; (2) the slow transition observed in bulk population analysis arises not from gradual puncta formation but from heterogeneity in lag times across individual cells. Importantly, the lag time itself is reliably determined, with a median of 2.2 h and an interquartile range of 1.1 to 3.0 h, well above our temporal resolution.

We agree that the limited resolution in capturing transition times may obscure potential correlations between transition and lag times. In response, we have now revised the text and clarify this point regarding Fig 2f as below (page 6, line 27-29):

...; we note that this lack of correlation could partly result from the limited temporal resolution of our imaging in resolving the shorter transition times....

- Pag. 8 line 20-27. The author rightfully dedicated care to the validation of the CymAPmC distribution, but no images of the CymAGFP are present. Especially in figure 3 it would be relevant to see both channels for all the presented condition and not only for the pair a-b and e-f.

[Reply 2-13] Thank you for your appreciation of our validation of CymA^{PmC} distribution in the manuscript. In addition, CymA^{GFP} localization was actually shown in both Fig. 3b and Supplementary Fig. 23 (now Supplementary Fig. 29) in our original submission, as part of our efforts to validate the distribution of CymA.

Moreover, regarding Fig. 3: panels a and b display the two fluorescence channels of the double-tagged CymA^{GFP} (green channel) + STC^{PmC} (red channel) strain, providing a direct comparison of the two proteins. Panels c and d originally represented CymA^{PmC} or STC^{PmC} expression, each in a single-tagged strain under H₂/fumarate growth condition, derived from two separate experiments. In response to your suggestion, we have now included two-channel imaging using the doubly-tagged strain (CymA^{GFP} + STC^{PmC}) for this condition to enhance consistency and clarity across the figure, as shown below. Panels e and f already show both channels for the double-tagged strain (CymA^{GFP} + FccA^{PmC}), as noted. For the $\Delta fccA$ deletion strain shown in panel g, only CymA^{GFP} is visualized, as the second protein FccA is not present. We hope these clarifications and the additional imaging data address your concerns and improve the completeness of the figure presentation.

Fig. 3 | CymA punctum formation is spatiotemporally coordinated with its direct EET partners. **a,b**, Fluorescence images of ensemble-photoactivated STC^{PmC} (**a**, red channel) and $CymA^{GFP}$ (**b**, green channel) in the same cells grown anaerobically with lactate/ferric citrate as the electron donor/acceptor pair for outflow EET. Same growth condition for (**e-g**). Inset: overlaid zoom-in image of STC^{PmC} and $CymA^{GFP}$ in dotted boxes. **c,d**, Fluorescence images of ensemble-photoactivated STC^{PmC} (**c**) or $CymA^{GFP}$ (**d**) in cells grown anaerobically with hydrogen/fumarate as the electron donor/acceptor pair where EET does not involve STC. **e,f**, Fluorescence images of ensemble-photoactivated $FccA^{PmC}$ (**e**, red channel) and $CymA^{GFP}$ (**f**, green channel) in the same cells. Inset: overlaid zoom-in image of $FccA^{PmC}$ and $CymA^{GFP}$ in dotted boxes. **g**, Fluorescence image of $CymA^{GFP}$ in $\Delta fccA$ cells. All scale bars: 5 μm .

- Pag. 10 line 29 and Supplementary Note 9 Fig 10. The concentration of the protein is often reported as an important parameter in the assay. For the partner proteins, there is more protein concentration in anaerobic conditions independently of the presence or absence of the condensate, and at the same time, the condensate correlates with higher photocurrent. How does CymA copy number vary in this picture?

[Reply 2-14] As shown in Figure 1e and mentioned in the manuscript in page 3, line 37-41, there is no significant change in CymA copy number. CymA appears to be constitutively expressed ($\sim 5.3 \mu M$ in concentration, $\sim 4.0 \times 10^3$ in copy number, on average), consistent with previous transcriptional analysis and showing no significant changes regardless of its active participation in EET or which external redox pairs were used.

- Instead of expressions like “first-of-its-kind” or “first”, it would be more valuable to directly address the status of the field until the presented work and how the approach introduces an important new observation.

[Reply 2- 15] We appreciate your suggestion. We have removed the phrase “first-of-its-kind” from the manuscript. We noted how our approach, correlated photocurrent measurement with fluorescence microscopy, directly informs on the functional roles of condensate formation in the physiological environment.

- All the paper is based on the single-cell quantification of protein concentration, although based on previous literature, I would appreciate a clear explanation of the method, even just in the supplementary, explaining the limitations of such quantification. Furthermore, is this the first time the author used PAmCherry for this assay? How does the different photophysics of the protein enter the quantification? How does bleaching enter into this estimation? For example, if the diffuse population falls below the SNR level faster than the aggregate.

[Reply 2-16] Thank you for pointing out to include the limitation of quantification method. We acknowledge there is some limitation of protein quantification method.

First, considering that our single-cell quantification of protein concentration is based on measuring overall cell fluorescence intensity and detecting single-molecule fluorescence in single cell, the approach is sensitive to background fluorescence of the cell, which can lead to false-positive signals. To address this question, we performed control imaging experiments using the same imaging process on a wild-type *S. oneidensis* strain that does not contain any fluorescent protein

tag. Some false positive single-molecule signals appeared during SMT, and a small fluorescence level was also present during SCQPC, together amounting to an apparent, false-detected copy number of 52 ± 36 molecules on average in each cell (new Supplementary Fig. 2). Compared with the copy number of CymA^{PmC} ($\sim 4.0 \times 10^3$ in copy number on average, see Supplementary Table 4 for details), this false-detection is less than 2% and thus negligible. For extremely low-copy number proteins, such artifacts would be problematic and need more experimental imaging optimization and correction, but the proteins we studied here have substantially higher copy numbers.

We have added the following paragraph and control results as Supplementary Fig. 2 in Supplementary Methods:

We performed the same SMT+SCQPC imaging process using a wild-type *S. oneidensis* strain that does not contain any fluorescent protein tag as a control experiment. There are some false positive single-molecule signals during SMT and a slight fluorescence level during SCQPC, amounting together to 52 ± 36 molecules on average per cell (Supplementary Fig. 2). Compared with the copy number of CymA^{PmC} ($\sim 4.0 \times 10^3$ in copy number on average, see Supplementary Table 4 for details), these false detections are less than 2% and thus negligible.

Supplementary Fig. 2 | Control SMT+SCQPC imaging for false detection of protein copy numbers in single cells. Copy number is obtained from the wild-type *S. oneidensis* that contains no fluorescent protein tags. 345 cells were counted. Mean \pm s.d. = 52 ± 36 .

Second, fluorophore maturation, photoactivation efficiency, and susceptibility to photobleaching can affect the accuracy of protein quantification. In our study, we confirmed the fluorophore is sufficiently matured before imaging (Supplementary Note 3), and photoactivation efficiency (i.e., 50% for PAmC1) is taken into account. For fluorescence bleaching, it does not play a role here because we used the first image frame to calculate the overall cell fluorescence intensity. While our imaging conditions were optimized to ensure consistent fluorophore performance and minimize photobleaching, we agree these sources of variability are intrinsic to fluorescence-based quantification and should be considered for similar protein quantification approaches. Per your suggestion, we mentioned such factors that need to be considered to avoid over/underestimation of protein quantification in Supplementary Methods 1.5.

Regarding the usage of PAmCherry1, it is not our first time using it for single-molecule/single-cell imaging. We have used it in imaging/tracking/quantifying cell envelope proteins in *E. coli* (Zhang et al. Transporter excess and clustering facilitate adaptor-protein shuttling for bacterial efflux. *Cell Rep. Phys. Sci.* **6**, 102441 (2025)), as well as hydrogenases and PHB granule associated proteins in *Ralstonia Eutropha* (Ref 38: Fu et al. Single-cell multimodal imaging uncovers energy conversion pathways in biohybrids. *Nat. Chem.* **15**, 1400–1407 (2023)). PAmCherry1 has also been used extensively in the literature for super-resolution imaging and single-molecule tracking as well (e.g., Subach et al. Photoactivatable mCherry for high-resolution two-color fluorescence microscopy *Nat. methods* **6**, 153-159 (2009); Thrall et al. Single-molecule imaging reveals multiple pathways for the recruitment of translesion polymerases after DNA damage. *Nat Commun* **8**, 2170 (2017)).

Minor comments

- Pag. 2 line 29-31. The sentence misses the verb, and the meaning is not clear.

[Reply 2-17] We revised the manuscript to make it clearer, on page 2, line 29-31.

... However, their spatiotemporal coordination is critical for biological electron transfer, which relies on the close proximity of electron-transfer partners, typically achieved through specific molecular interactions²¹. ...

- Supplementary fig 25 is a repetition of figure 4d.

[Reply 2-18] They may appear similar but the two figures present different data. Fig 4d presents the correlation between CymA concentration and photocurrent, whereas Supplementary Fig. 25 (now Supplementary Fig. 13) presents the correlation between CymA concentration and photocurrent *density*. We have mentioned their differences in the manuscript; The caption of this Supplementary Figure also describes the difference from Fig. 4d in more detail.

Reviewer #2 (Remarks on code availability):

The code is provided and commented in its components, but the inclusion of example datasets would be essential for accessibility and practical use.

[Reply 2-19] Thank you for catching this omission. We now added example datasets in the supplementary code and the directory pathways are revised.

[To all reviewers] Thank you again for reviewing our manuscript and for your comments. We have revised accordingly, including point-by-point replies below. All revisions in the main text and SI are marked in red fonts and cross-cited here. The revised text reproduced from the manuscript and SI in the replies below is also in red fonts.

We also want to apologize for the longer turnaround than we hoped in preparing this round of revision. The 1st author, Youngchan Park, has moved to Indiana University to start her independent career, busy with setting up her lab while teaching new courses. Thank you.

Reviewer #1 (Remarks to the Author):

The authors have provided logical and appropriate responses to the concerns I raised in the previous round of review. While some of the experimental issues could not be addressed with additional data, the authors have thoroughly explained the technical limitations and convincingly argued that the lack of these experiments does not undermine the main conclusions. I recommend acceptance of the manuscript after minor revisions.

[Reply 1-0] Thank you very much for your recognition of our efforts to address previous concerns. We have carefully implemented the suggested minor revisions to further strengthen the manuscript.

Comment on Reply 1–2:

In response to my previous concern that “Aggregate formation of FccA/STC could hinder efficient diffusion in the periplasmic space due to the increased protein size and the reduced number of FccA/STC available for collision. Please discuss the potential disadvantages of aggregate formation on the efficient EET in the discussion part,” the authors replied that “This dynamic association/dissociation of STC/FccA with CymA, coupled with the rapid diffusion of free periplasmic components, should allow efficient ET between CymA on the inner membrane and the Mtr complex on the outer membrane, mediated by FccA and STC.”

I appreciate the authors’ effort to clarify that the FccA/STC puncta do not represent static aggregates but rather dynamic assemblies undergoing continuous association and dissociation with CymA. This addresses my initial assumption and adds valuable context to the functional interpretation.

However, I remain concerned about this explanation. The manuscript’s interpretation of FccA/STC colocalization with CymA is understandable if one assumes there are no intermolecular interactions among FccA or STC themselves. However, there is no clear experimental evidence supporting the absence of such interactions during puncta formation. It would be rather reasonable to assume that FccA/STC has some interaction, as FccA/STC is condensed, similar to CymA, which potentially exchanges electrons among CymAs, as the author mentioned. Therefore, it remains reasonable to consider that punctum formation could have a negative impact on EET efficiency.

Moreover, in Figure S34, the authors additionally stated that “(v) STC and FccA can subsequently dissociate from the condensate and diffuse toward the outer-membrane to exchange electrons with the MtrABC/OmcA complex.” If it were shown that reduced FccA/STC molecules dissociate from the puncta following electron uptake, it would support the proposed model. On the contrary, the authors suggest that the reduction reaction possibly triggers FccA/STC punctum formation rather than dissociation, as supported by the additional experiment demonstrating that electron donation from CymA to STC is necessary for colocalization of STC with CymA mentioning that “Remarkably, STCPmC maintains a homogenous spatial distribution (Fig. 3c), while CymAGFP still formed puncta (Fig. 3d)”. This, I think, further undermines the claim that these proteins can readily dissociate and diffuse to the outer membrane.

Additionally, the authors have added the following in the main text that “Thus, while FccA and STC may not independently form condensates or mix into CymA condensates, their spatial proximity to CymA puncta is mediated through direct protein-protein interactions. This interaction-driven colocalization effectively brings the electron transfer partners into proximity.” While this explanation is plausible and consistent with known protein-protein interactions, the specific novelty of this statement is somewhat difficult to discern. If the main point is that direct interactions between FccA/STC and CymA contribute to efficient EET, this mechanism has already been well established in previous studies. As the authors themselves acknowledge, complex formation between these components is a canonical feature of the *Shewanella* EET pathway. Therefore, if the punctum formation does not involve additional intermolecular interactions among FccA or STC themselves, it remains somewhat unclear how puncta formation contributes to enhanced geometric proximity or improved electron transfer efficiency. Clarifying whether and how the spatial organization introduced by puncta formation provides functional advantages would help better establish the physiological relevance of this phenomenon.

[Reply 1-1] Thank you for your further comments.

First, we would like to clarify that FccA/STC colocalization with CymA condensates does not necessarily imply that a larger fraction of FccA/STC molecules participate in EET. As you noted, interactions between FccA/STC and CymA are

well established; however, it remains unclear whether these interactions differ under aerobic versus anaerobic growth conditions. If the interactions occur independently of growth condition, then regardless of condensate formation, a fraction of FccA/STC would always form complexes with IM-anchored CymA, while the rest fraction would remain free in the periplasm. In this case, the proportion of free FccA/STC molecules available for collision with Mtr complex would remain similar. Our data show that anaerobic growth conditions led to CymA condensate formation and thereby increase FccA/STC colocalization at localized puncta, and *at the same time*, that the copy numbers of FccA and STC themselves increase markedly under these conditions (by ~300% and ~70%, respectively; Supplementary Fig. 28), while CymA's copy number remains similar. Thus, under anaerobic growth, the absolute number of free FccA/STC molecules available for collision with Mtr complex would increase. Therefore, we do not consider colocalization of FccA/STC with CymA to hinder diffusion or to represent a disadvantage for efficient EET.

We agree that there is no direct experimental evidence demonstrating the absence of intermolecular interactions among FccA/STC during puncta formation; however, there is also no evidence supporting their presence, even though FccA/STC are spatially colocalized/condensed via interactions with CymA. Since these possibilities remain highly speculative, we believe that arguing for one or the other in the discussion could be misleading and cause unnecessary confusion for future research. Therefore, we prefer to focus on solid conclusions that can be drawn from our experimental findings; we hope you would agree with our being cautious.

To address your concern on the dynamic association/dissociation, we have added single-molecule tracking trajectories of STC/FccA moving into condensate, and some of moving out of condensate, in a new Supplementary Fig. 37, and cross-cited it in the main manuscript on page 15, line 5 and in Supplementary Fig. 34 caption.

Supplementary Fig. 37 | Dynamic interactions of STC/FccA with CymA condensates. **a,b**, Representative single-molecule tracking trajectories (colored lines) overlaid on wide-field fluorescence images of ensemble-photoactivated (a) STC^{PmC} and (b) FccA^{PmC}. These single-molecule trajectories show periplasmic proteins moving into and out of condensates, supporting dynamic association and dissociation of STC/FccA with CymA condensate. Dashed yellow circle: condensate outline. Scale bars: 1 μm .

Other than known to be periplasmic or inner-membrane proteins, the 'spatial' behaviors, especially condition-/growth-dependent spatial behaviors, of FccA/STC and CymA during EET were not known previously, even though they were known to interact for electron exchange. Our results show that significant fractions of FccA/STC colocalize into condensed regions following CymA puncta formation, which is a novel finding here. We propose that the entire sequence, from CymA puncta formation to the subsequent spatial colocalization of periplasmic proteins, enhances EET efficiency: (1) a primary functional advantage of CymA condensate formation lies in facilitating electron exchange with menaquinone within the condensate in the inner membrane. (2) The colocalization of FccA/STC with CymA in the condensate also has a potential functional advantage: if the original CymA protein that a FccA/STC protein interacts with is not in the proper redox state for electron exchange, the colocalization within the condensate will facilitate the FccA/STC protein to sample nearby other CymA proteins for electron exchange.

We have revised on page 14, line 30 as follows:

“These CymA puncta are likely biomolecular condensates associated with IM domains enriched with the anaerobic electron carrier menaquinone, which should facilitate the electron exchange between CymA and menaquinone. **The dynamic mobility of CymA within these condensates as well as the colocalization and dynamic interactions with STC/FccA may also enable STC/FccA to sample multiple CymA molecules to find those with suitable redox states**

for effective electron exchange before STC/FccA molecules diffuse toward EET proteins on the OM (Supplementary Fig. 37).”

Comment on Reply 1–4:

The authors added the following sentence that “Most CymAPmC puncta were observed in cells within 4 h during which there was no substantial new synthesis of CymAPmC in the cell (Supplementary Notes 3).” However, I was unable to confirm this statement from Supplementary Notes 3. While it is clear that CymAPmC concentrations did not increase substantially within 4 hours, the data do not provide direct evidence that most puncta were observed within this 4-hour window. If the authors wish to make this claim, they should provide data on the number of puncta per cell over time in Supplementary Notes 3 to substantiate it.

[Reply 1-2] Apologies for the confusion. In Figure 2f, 41 out of 45 cells formed condensates within 4 hours, while only 4 cells did so after 4 h. Thus, over 90% cells exhibited puncta formation within 4 hours. We have revised as follows (page 6, line 2):

“Most CymA^{PmC} puncta were observed in cells within 4 h (Fig. 2f, >90%) during which there was no substantial new synthesis of CymA^{PmC} in the cell (Supplementary Notes 3)”

Reviewer #1 (Remarks on code availability):

As mentioned, I did not review the code, because there were no comments and replies related to the code for this revision.

Reviewer #2 (Remarks to the Author):

I appreciate the additional clarifications and new data provided. Some points are now clearer, but I still have some questions that could further improve clarity and depth of the work.

[Reply 2-0] Thank you very much for reviewing our manuscript and appreciating our efforts to address your previous comments, which helped us improve greatly.

Reversibility.

- It remains unclear why the reversible nature of the puncta could not be examined using the single-cell assay. From Supplementary Figure 25, if the potential is interrupted, would the puncta dissipate or behave differently? By reducing imaging frequency, could the observation be extended, especially under the faster condition tracked in Figure 2?

[Reply 2-1] In our last-round response **[Reply 2-2]**, we discussed technical difficulties on tracking the same set of cells over time while switching aerobic and anaerobic solutions for testing reversibility. With respect to puncta reversibility under electrochemical manipulation, our data already demonstrate that these puncta emerge and dissipate under constant potential (Fig. 2d).

We also agree that after puncta formation, interruption of potential could in principle lead to puncta dissipation. But unfortunately, we have not done such an experiment. We have practical difficulties in fulfilling these experiments currently: The 1st-author, Youngchan Park, has left the group and started her independent career at Indiana University, busy setting up her own lab and teaching new courses. Optimizing the imaging condition for such an experiment would take extensive efforts beyond our current practical capability, and perhaps should be left for the future. Nevertheless, we believe the manuscript already presents sufficient evidence supporting the reversible nature of CymA puncta, including: (i) temporal emergence and dissipation of puncta upon applying potential (Fig. 2d) and (ii) reversible redistribution in response to growth condition changes (Supplementary Fig. 36). We very much hope you would agree that these observations are sufficient to support the reversibility of CymA puncta formation.

- In Supplementary Figure 35, membrane localization is very clear in the control (panels a–c) but much less so in the “return” condition (panels g–i). Is this variability due to the specific example shown, or is it a general feature of CymA redistribution? Please clarify this variability in protein spatial distribution. Also in many images with puncta, membrane localization appears less clear. Adding a line profile orthogonal to the bacterium could help highlight and quantify this difference.

[Reply 2-2] Thank you for pointing this out. We agree that in some cases membrane localization is less evident in the images, which are 2D projections. As you suggested, we have added representative intensity line profiles in panels d, h, and l of Supplementary Fig. 36 (previously Supplementary Fig. 35) and comment on this observation in the caption to clarify the variability in protein spatial distribution.

Supplementary Fig. 36 | ... (d) Fluorescence intensity profiles along the line marked by a white dashed arrow in (c). ... (h) Fluorescence intensity profiles along the line marked by a white dashed arrow in (g). ... (l) Fluorescence intensity profiles along the line marked by a white dashed arrow in (k). Note that we observed that membrane localization of CymA somewhat decreased after returning to aerobic condition...

Spatial Description.

• On page 3, line 42, the term “spatial heterogeneity” remains ambiguous—does this refer to the percentage of cells with puncta, their shape, location, or size?

[Reply 2-3] There may be some misunderstanding here, as the term “spatial heterogeneity” was not used in our manuscript or SI. The sentence on page 3, line 42 states: “Moreover, our imaging unveiled heterogeneity in punctum formation among individual cells (Fig. 1c–d and Supplementary Fig. 21).” Here, we described heterogeneity in punctum formation across cells, i.e., some cells already formed puncta while others had not.

• The updated size quantification appears coarse; the threshold-based approach may obscure nanoscale features and limits assessment beyond the diffraction limit. While I understand this may be outside the main scope of the study, Supplementary Figure 6 raises further questions. In this figure, membrane localization of the clusters is difficult to discern. Given the clear membrane localization observed in the control widefield images, wouldn’t the spatial resolution employed be expected to reveal a membrane outline? Please also clarify the effective spatial resolution of your imaging system, particularly for single-molecule tracking.

[Reply 2-4] It is true that membrane localization of the clusters is a bit difficult to discern in cells that formed puncta. In the super-resolution images, the membrane outline appears less clear because only a small fraction of fluorescent protein is present outside the puncta, and super-localization relies on the density of localized molecules. By contrast, in the control

widefield images (aerobic condition, no puncta), the proteins are more evenly distributed across the membrane, which makes the outline more visible. The effective localization uncertainty of our single-molecule imaging is ~ 40 nm and we added the statement in Supplementary Methods 1.5.

Dynamic profile.

- The FRAP data support the idea of biomolecular condensates, but currently appear as single, isolated examples. A broader, statistically representative dataset would strengthen this evidence.

[Reply 2-5] Thank you for suggesting FRAP experiment to further support our claim that CymA molecules are dynamic in puncta, consistent with features associated with biomolecular condensates. In response, we have included additional FRAP curves in Supplementary Fig. 32.

Supplementary Fig. 32 | Fluorescence recovery after photobleaching (FRAP) measurement. a... b, Three normalized fluorescence intensity trajectories, fitted with exponential function. **The data in red line is the data shown in (a).** The fluorescence in the bleached region recovered with a half time of 3.4 ± 2.2 min to approximately $76 \pm 52\%$ of the initial intensity at steady state, indicating that CymA molecules in puncta are mobile. **The other two bleached regions recovered with half-times of 4.4 ± 4.2 min (upper black line) and 4.3 ± 2.4 min (lower black line), reaching $\sim 123 \pm 73\%$ and $\sim 32 \pm 11\%$ of their initial intensities, respectively. All errors are 95% confidence bounds from fits. ...**

We acknowledge that, even with these additions, the dataset may remain statistically limited. However, we believe that we already have presented sufficient data to show that CymA molecules are dynamic, including single-molecule tracking (Fig. 5a, Supplementary Note 12), CymA molecules' in-and-out kinetics (Fig. 5g, Supplementary Note 13.2, and Supplementary Note 14), temporal emergence and dissipation of puncta (Fig. 2d), and puncta dissipation after returning to aerobic condition (Supplementary Fig. 36). Together, these observations reinforce our interpretation.

Unfortunately, due to the 1st-author's departure to an independent position, it has not been feasible to collect further FRAP data at this stage. We very much hope you would find the additional analyses and complementary evidence sufficient to support our conclusions.

- In Figure 3, the new data improve readability, but given the heterogeneous signal (some cells are very bright in the "red" channel, while other extremely dim), adding line profiles across cells could help confirm puncta presence and protein accumulation.

[Reply 2-6] Thank you for the suggestion. We have added line profiles across cells in Fig 3c and d, and present in the new Supplementary Fig. 29.

Supplementary Fig. 29 | Intensity line profile analysis shows clearly that STC is homogeneously distributed when not involved in EET, while CymA forms puncta. **a,b**, Fluorescence images of ensemble-photoactivated STC^{PmC} (a) or CymA^{GFP} (b) in cells grown anaerobically with hydrogen/fumarate as the electron donor/acceptor pair where EET does not involve STC. These figures are replicate of Fig. 3 c-d in the manuscript. Scale bar = 5 μm. **c, d**, Zoom-in image of the yellow box (i) and blue box (ii) in a-b. White arrows denote the direction of line drawing. Normalized intensity line profile of STC^{PmC} (red line) showed apparent bipolar accumulation due to homogeneous membrane localization, while that of CymA^{GFP} (green line) showed much higher intensity at one end due to puncta formation. These data clearly support that CymA punctum formation can occur independently from STC, indicating that STC is not the driver of CymA punctum formation.

Minor points.

- The disposition of panel in figure 5 is confusing, considering the sudden change respect to the previous figures.

[Reply 2-7] We understand the concern; however, we believe that the current arrangement allows for a clearer comparison among panels c, d, e, and f.